# An image-computable model for the stimulus selectivity of gamma oscillations

**Dora Hermes[1,2]\*, Natalia Petridou[3], Kendrick N Kay[4†], Jonathan Winawer[5,6†]**

[1]Department of Physiology and Biomedical Engineering, Mayo Clinic, Rochester, United States; [2]Department of Neurology and Neurosurgery, UMC Utrecht Brain Center, Utrecht, Netherlands; [3]Center for Image Sciences, University Medical Center Utrecht, Utrecht, Netherlands; [4]Center for Magnetic Resonance Research (CMRR), Department of Radiology, University of Minnesota, Minneapolis, United States; [5]Department of Psychology, New York University, New York, United States; [6]Center for Neural Science, New York University, New York, United States

**\*For correspondence:**
hermes.dora@mayo.edu

[†]These authors contributed equally to this work

**Abstract** Gamma oscillations in visual cortex have been hypothesized to be critical for perception, cognition, and information transfer. However, observations of these oscillations in visual cortex vary widely; some studies report little to no stimulus-induced narrowband gamma oscillations, others report oscillations for only some stimuli, and yet others report large oscillations for most stimuli. To better understand this signal, we developed a model that predicts gamma responses for arbitrary images and validated this model on electrocorticography (ECoG) data from human visual cortex. The model computes variance across the outputs of spatially pooled orientation channels, and accurately predicts gamma amplitude across 86 images. Gamma responses were large for a small subset of stimuli, differing dramatically from fMRI and ECoG broadband (non-oscillatory) responses. We propose that gamma oscillations in visual cortex serve as a biomarker of gain control rather than being a fundamental mechanism for communicating visual information.
DOI: https://doi.org/10.7554/eLife.47035.001

## Introduction

An important goal in visual neuroscience is to develop models that can predict neuronal responses to a wide range of stimuli. Such models are a test of our understanding of how the system functions, and have led to insights about canonical computations performed by the early visual system, such as filtering, rectification, and normalization (*Carandini et al., 2005*). Image-computable models, which predict responses to arbitrary, unlabeled images, have been developed for the functional MRI (fMRI) blood oxygen level dependent (BOLD) signal (*Dumoulin and Wandell, 2008*; *Güçlü and van Gerven, 2015*; *Kay et al., 2013b*; *Kay et al., 2013a*) and for spiking of single neurons in animals (*Mante et al., 2008*; *Rust et al., 2005*; *Simoncelli and Heeger, 1998*). In contrast, to our knowledge, there are no image-computable models to predict oscillations in the gamma band (30–80 Hz) of the local field potential (but see Discussion), although gamma oscillations have been intensely studied and are a prominent part of many neural recordings and theories of neural function (*Jensen et al., 2007*; *Tallon-Baudry and Bertrand, 1999*).

Studies in humans and animal models indicate that gamma oscillations are systematically related to image properties, supporting the possibility that this stimulus dependence might be captured in an image-computable model. In particular, oriented bars (*Gray and Singer, 1989*; *Gray et al., 1989*) and gratings (*Eckhorn et al., 1988*; *Hermes et al., 2017a*; *Muthukumaraswamy and Singh, 2008*) elicit large gamma responses in visual cortex. The responses tend to increase with stimulus size and contrast (*Gieselmann and Thiele, 2008*; *Henrie and Shapley, 2005*; *Ray and Maunsell,*

*2011*), and tend to decrease in the presence of unstructured noise (*Bartoli et al., 2019*; *Hermes et al., 2015*; *Jia et al., 2013*; *Kayser et al., 2003*) or multiple orientations (*Bartolo et al., 2011*; *Lima et al., 2010*). Some images of scenes and objects cause large reliable gamma oscillations while others do not (*Brunet et al., 2015*; *Hermes et al., 2015*). Gamma response selectivity differs from that of the BOLD response (*Hermes et al., 2017b*; *Muthukumaraswamy and Singh, 2009*) and single- and multi-unit spike rates (*Jia et al., 2013*; *Peter et al., 2019*; *Ray and Maunsell, 2011*). For example, when recording from the same electrode in primary visual cortex (V1), increasing grating size causes gamma oscillations to increase in power while causing firing rates to decrease (*Jia et al., 2013*; *Ray and Maunsell, 2011*). Gamma power also decreases dramatically with noise masking while population firing rates do not change substantially (*Jia et al., 2013*). Because image selectivity in gamma oscillations clearly differs from the selectivity in firing rates and BOLD signals, a model to predict the extent to which different images will give rise to gamma oscillations requires a different form than a model to predict the BOLD signal or firing rates.

Here, we measured responses from electrocorticography (ECoG) electrodes over visual cortex while human subjects viewed a variety of different images. We separated the ECoG response into two spectrally overlapping components: one broadband (spanning 30–200 Hz) and one narrowband (centered between 30–80 Hz). We compared the broadband component and narrowband gamma component to the images, and developed image-computable models to account for the stimulus selectivity present in each. The broadband response was well fit by a model adapted from fMRI of visual cortex (*Kay et al., 2013b*). The narrowband gamma responses were strikingly different, and we developed a new, image-computable model in order to explain those responses. The differences in the patterns of responses and the differences in the two models suggest that broadband signals and narrowband gamma originate from distinct aspects of neural circuitry.

## Results

In order to develop an image-computable model that can predict gamma responses to a large variety of images, we measured ECoG signals in three human subjects. We identified ECoG electrodes that were located on the surface of V1, V2 and V3 and had a well-defined population receptive field (pRF) measured from an independent experiment with sweeping bar stimuli (as in *Winawer and Parvizi, 2016*; *Winawer et al., 2013*). This yielded six electrodes in the first subject, two in the second subject and seven in the third subject (*Figure 1A*). For each electrode we calculated the power spectra from the 500 ms window following presentation of each of the 86 images. As in previous work (*Hermes et al., 2015*), we separated the power spectra into an oscillatory and non-oscillatory component by modeling the log-power/log-frequency spectrum as the sum of 3 components: a linear baseline, a constant, and a Gaussian centered between 30 and 80 Hz (*Figure 1B*). These three terms correspond to the baseline signal in the absence of a stimulus, a stimulus-specific broadband response, and a stimulus-specific narrowband gamma response, respectively. In our quantification, the narrowband gamma and the broadband signal overlap in frequency content, and are distinguished by the pattern in the spectral data. Despite the overlap, the two components do not depend on one another: it is possible to obtain a positive broadband component with zero narrowband gamma, zero broadband component with positive narrowband gamma, or a mixture of the two types of components. Note that our quantification of narrowband gamma response was constrained to be a nonnegative Gaussian, and so in the presence of noise, even images which cause the oscillatory gamma response to decrease or to remain unchanged (compared to the blank screen) are likely to be estimated as slightly positive.

Replicating previous results (*Hermes et al., 2015*; *Jia et al., 2013*; *Zhou et al., 2008*), we observe large increases in narrowband gamma power for large, high-contrast grating patterns but not for noise patterns, and strong broadband responses for both grating patterns and noise stimuli (*Figure 1B–C*). Because the broadband response is similar for the two types of stimuli, the greatly reduced narrowband gamma for the noise stimuli does not indicate a general lack of visual responsivity, but rather that a different type of stimulus selectivity exists for the gamma response compared to the broadband response.

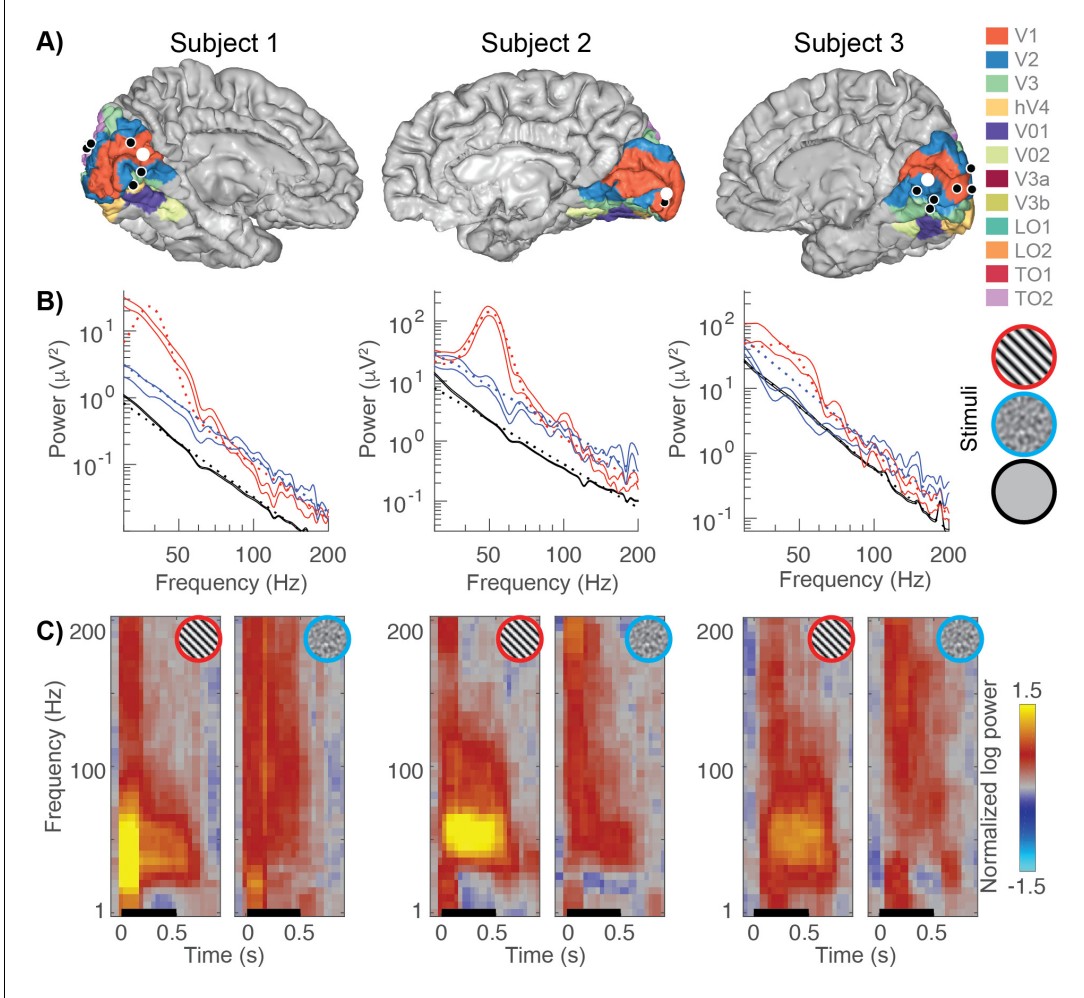

**Figure 1.** Example broadband and gamma responses to grating and noise stimuli. (A) Location of the electrodes implanted in each subject (black and white dots) rendered on estimates of early visual areas (*Benson and Winawer, 2018*; *Benson et al., 2012*). (B) The power spectra for example electrodes (white dots from panel A). Power spectra are shown on a double logarithmic plot for a grating stimulus (red, stimulus number 45), a noise pattern (blue, stimulus number 83) and the baseline condition (black). The solid lines indicate the data (68% confidence interval from bootstrapping). The dotted lines indicate the fits to the data: stimulus-induced responses are modeled as a baseline linear fit (black) plus a constant and Gaussian to capture broadband and narrowband stimulus-specific responses, respectively. (C) Time-frequency plots (spectrograms) for the same electrodes. The black line indicates stimulus timing (500 ms). All spectrograms are normalized with respect to the same baseline: the inter-stimulus interval between all trials (from 250 to 500 ms after stimulus offset). Spectrograms are cut off at a maximum power of ±1.5 log10 units. The multitaper approach results in a temporal smoothing of 200 ms and a frequency smoothing of ±15 Hz. Spectrograms represent averages across all trials of a given stimulus type. Code to reproduce this figure can be found on GitHub (*Hermes, 2019*).

DOI: https://doi.org/10.7554/eLife.47035.002

The following figure supplement is available for figure 1:

**Figure supplement 1.** Phase locking plots.

DOI: https://doi.org/10.7554/eLife.47035.003

## Stimulus selectivity differs for ECoG gamma oscillations compared to broadband ECoG and BOLD

We find that stimulus selectivity is similar for broadband ECoG and BOLD but quite different for gamma oscillations. We first illustrate this with example responses from the three signal types. BOLD responses to a subset of these images were measured in previous work (*Kay et al., 2013b*) (images 1 to 77, *Figure 2A*). Using the publicly available data from that study (http://kendrickkay.net/socmodel/), we identified a voxel in V1 with the most similar pRF location to one of our electrode's pRF (electrode 3 in subject 1), and plotted the BOLD responses from this voxel to the different

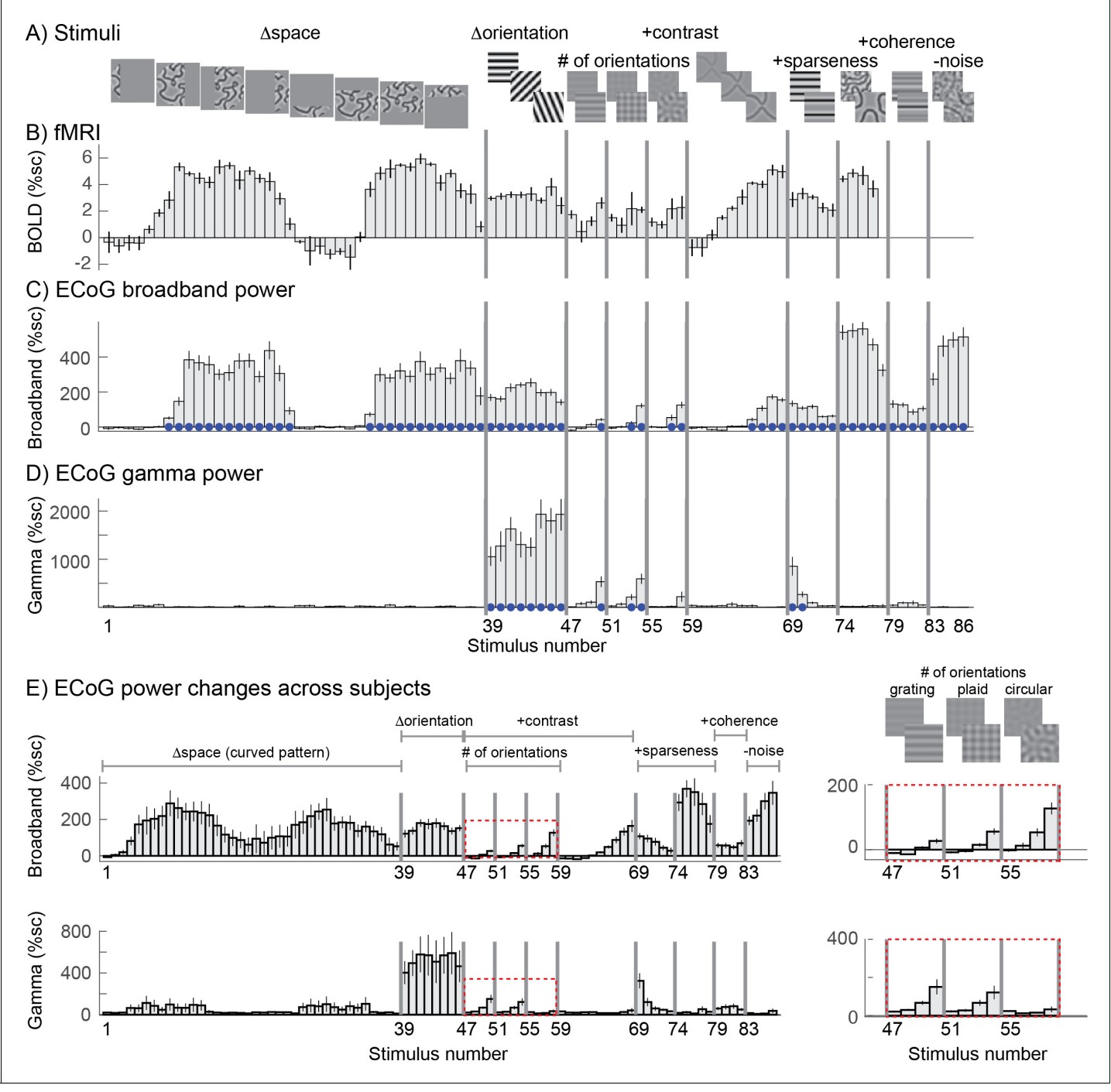

**Figure 2.** Stimulus selectivity for BOLD, broadband, and gamma. Responses in human V1 to different images expressed as percent signal change above baseline. (**A**) The images were identical to those used in a prior fMRI study, with a few additions, and are grouped into several stimulus categories: SPACE (1 to 38), ORIENTATION (39 to 46), CONTRAST (47 to 68), SPARSITY (69 to 78), COHERENCE (79 to 86). Larger versions of the stimuli are shown in *Figure 2—figure supplement 1–5*. (**B**) The fMRI BOLD response in one V1 voxel whose pRF location is matched to the electrode shown in C and D (replotted from *Kay et al., 2013b*). (**C**) The ECoG broadband response for all 86 images from a single electrode (*Figure 1A*, left). (**D**) The ECoG narrowband gamma response for all 86 images, recorded in the same electrode. (**E**) Broadband (top) and narrowband (bottom) gamma responses to 86 stimuli averaged across the 15 electrodes in V1-V3. Insets (right) show a zoomed-in view of how the responses vary with four contrast levels and different numbers of component gratings: 1, 'grating', 2 'plaid', or 16 'circular'. Within each type of pattern, the four bars are responses to stimuli with increasing stimulus contrast. In panels C-D, blue dots indicate significant responses (p<0.05 by bootstrap). In panels B-D, error bars represent the 68% range by bootstrapping across trials. In panel E, error bars represent the 68% range by bootstrapping across electrodes. Code to reproduce this figure can be found on GitHub (*Hermes, 2019*).

*Figure 2 continued on next page*

*Figure 2 continued*

DOI: https://doi.org/10.7554/eLife.47035.004

The following figure supplements are available for figure 2:

**Figure supplement 1.** Stimuli 1-20.

DOI: https://doi.org/10.7554/eLife.47035.005

**Figure supplement 2.** Stimuli 21-40.

DOI: https://doi.org/10.7554/eLife.47035.006

**Figure supplement 3.** Stimuli 41-60.

DOI: https://doi.org/10.7554/eLife.47035.007

**Figure supplement 4.** Stimuli 61-80.

DOI: https://doi.org/10.7554/eLife.47035.008

**Figure supplement 5.** Stimuli 81-86.

DOI: https://doi.org/10.7554/eLife.47035.009

stimuli (*Figure 2B*). The BOLD responses in this voxel across stimuli are representative of the BOLD data reported in our previous study (cf. Figure 7 in *Kay et al., 2013b*). The pattern of responses in the BOLD data is generally similar to that of the broadband ECoG data (*Figure 2C*). In contrast, the ECoG narrowband gamma responses (*Figure 2D*) are quite different. The most salient pattern across the 3 types of responses to the whole image set is substantial BOLD and broadband responses for most images whereas gamma responses are large only for gratings (Δ*orientation*). The BOLD signal and ECoG broadband power are largest for patterns with multiple orientations, such as the curvy patterns in images 74-78 (and similar patterns from the Δ*space* stimuli that overlapped the pRFs, including images 8-16 and 30-36). These responses often exceeded the response to high contrast gratings (images 39-46). The opposite is true for the gamma responses, for which the biggest response by far is to high-contrast gratings, and for which no significant responses were found for space stimuli (relative to the response to blank stimuli, based on bootstrapping across trials; see Materials and methods). For all three signal types, if there is a response to a stimulus of a particular pattern, such as gratings (images 47-50) or plaids (images 51-54), then responses increased with stimulus contrast for that pattern. Note that the scale of the different measures varies considerably (e.g.,~5% signal change for BOLD, 200% for broadband, 2,000% for gamma) and are not comparable. We return to this issue in the section 'Gamma responses are well predicted by a model that is sensitive to variation in orientation content'.

These patterns are clearly evident not just in the example electrode (*Figure 2C–D*), but also in the responses averaged across the 15 electrodes in V1-V3 of 3 subjects (*Figure 2E*). In particular the broadband responses to curved patterns (stimuli 1–39) are high whereas the gamma responses are relatively low compared to the grating responses. Another systematic difference between the two types of responses is that broadband increases with the number of component gratings, whereas narrowband gamma decreases with the number of component gratings (*Figure 2E* insets, showing the responses to stimuli made from 1 (grating), 2 (plaid), or 16 (circular) component gratings).

## Broadband changes are well predicted by a model developed for fMRI

A variety of models have been developed to predict visually evoked fMRI signals, ranging from simple linear isotropic pRF models (*Dumoulin and Wandell, 2008*) to high-dimensional filter models with thousands of basis functions (*Eickenberg et al., 2017*; *Güçlü and van Gerven, 2015*; *Kay et al., 2008*) to cascade models composed of a small number of canonical computations (*Kay et al., 2013b*). (For a review see [*Wandell and Winawer, 2015*]). Since ECoG broadband responses typically correlate well with BOLD in visual cortex (*Hermes et al., 2017b*; *Winawer et al., 2013*), we tested whether a second-order contrast model (SOC) developed for fMRI could accurately fit, and predict, the ECoG broadband signal (*Kay et al., 2013b*). The SOC model is a two-stage LNN (Linear-Nonlinear-Nonlinear) cascade model. The first stage includes filtering the images with oriented Gabor functions (L), rectification (N), and divisive normalization (N). The second stage includes spatial summation within a pRF (L), calculation of second-order contrast (N) and a compressive nonlinearity (N). *Figure 3* shows the ECoG broadband responses for two example electrodes in each of the three subjects measured in V1, V2 or V3 (all electrodes are shown in *Figure 3—figure supplements 1–2*). The SOC model was fit to the broadband changes, using leave-one-stimulus-out

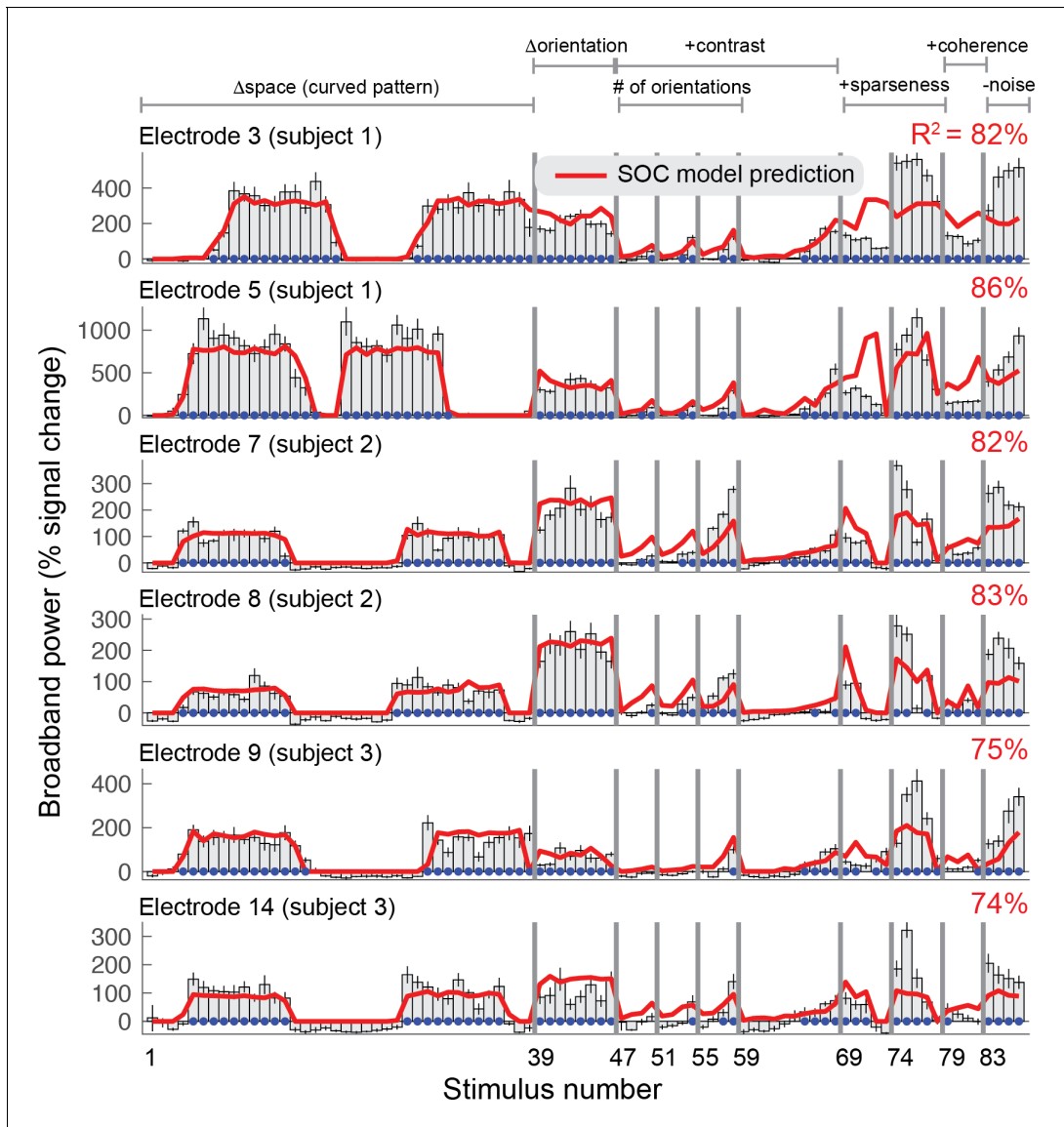

**Figure 3.** Second-order contrast (SOC) model accounts for ECoG broadband responses. Each row shows the percent signal change in ECoG broadband power for all 86 stimuli for six of the 15 electrodes on V1, V2 or V3. Error bars display the 68% range for bootstrapped responses (bootstrapped across repeated presentations of the same stimuli). The SOC model was fit to these data using leave-one-stimulus-out cross-validation. The cross-validated predictions and amount of variance explained are shown in red. The blue dots indicate that the stimulus response was significantly greater than baseline (p<0.05, bootstrap test). Code to reproduce this figure can be found on GitHub (*Hermes, 2019*).
DOI: https://doi.org/10.7554/eLife.47035.010

The following figure supplements are available for figure 3:

**Figure supplement 1.** Second-order contrast (SOC) model accounts for ECoG broadband responses for electrode 1–8.
DOI: https://doi.org/10.7554/eLife.47035.011
**Figure supplement 2.** Second-order contrast (SOC) model accounts for ECoG broadband responses for electrodes 9–15.
DOI: https://doi.org/10.7554/eLife.47035.012
**Figure supplement 3.** Broadband power changes calculated for a time window after the evoked response.
DOI: https://doi.org/10.7554/eLife.47035.013

cross-validation to control for overfitting and to obtain unbiased estimates of model performance. Although there are some discrepancies (especially in stimuli 79–86), the model fit the data well overall and explained an average of 80% (coefficient of determination) of the cross-validated variance across the 15 ECoG electrodes.

# An image-computable model of narrowband gamma responses

The striking difference in the selectivity observed in gamma responses compared to BOLD and broadband motivated us to develop a new model. Whereas the SOC model predicts increased responses for images with multiple orientations (e.g., plaids compared to gratings) and for moderately sparse patterns compared to gratings, the gamma selectivity exhibits the opposite pattern. We propose a new model, called the Orientation Variance (OV) model, that computes variance across the outputs of spatially pooled orientation channels within a pRF. Similar to the SOC model, the OV model builds on known computations in the visual cortex, but differs in the way signals are pooled across orientation bands (*Figure 4*). In particular, the model first sums outputs across space within an orientation band. This computation can be thought of as an analog to the long-range horizontal

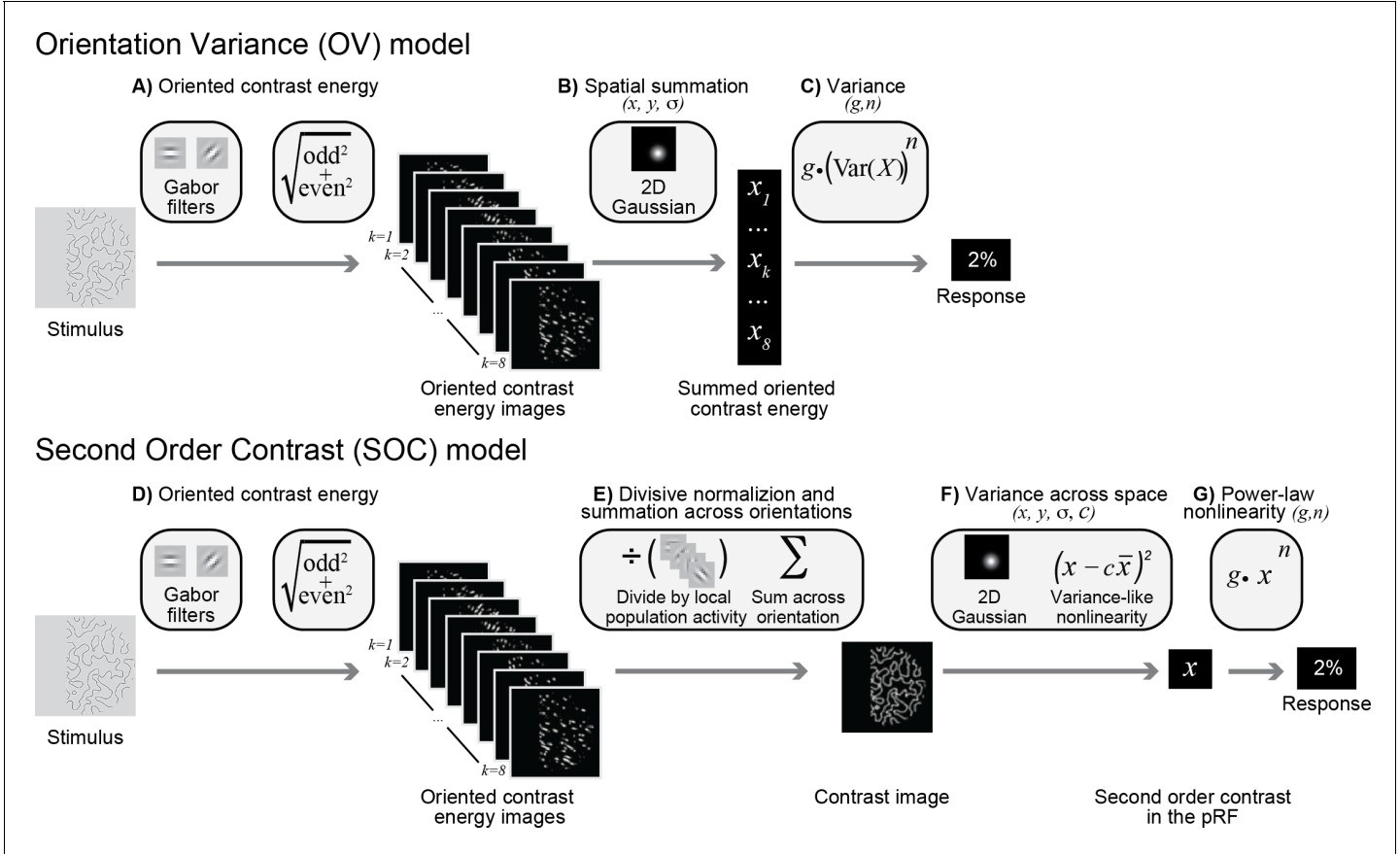

**Figure 4.** Orientation Variance (OV) model of gamma responses and the Second Order Contrast (SOC) model of broadband responses. Top: in the OV model, responses are largely driven by contrast and variance across orientations in the population receptive field. (A) Oriented contrast energy. Images are filtered with quadrature pair Gabor filters occurring at eight orientations. The quadrature pairs are summed across phase, resulting in eight images with contrast energy for each orientation. (B) The contrast energy within each of these eight images is summed within a population receptive field defined by a Gaussian with parameters $x$, $y$ and $\sigma$. This results in eight values, indicating the summed contrast energy within the pRF for each orientation. (C) Variance is calculated across these eight values, followed by a power-law nonlinearity ($n$) and a gain ($g$). Intuitively, the model predicts a large response when only one or a few orientations have high contrast energy and a low response when all orientations have similar contrast energy. Bottom: in the SOC model, responses are driven by contrast and variance across space in the population receptive field. (D) Contrast energy. The first step that generates oriented contrast energy images is the same as the OV model (see A). (E) Divisive normalization and summation across orientations. This results in a contrast image where information about variance across orientation is lost. (F) The image is then filtered by the population receptive field defined by a Gaussian with parameters $x$, $y$ and $\sigma$ and the variance in contrast compared to the mean contrast is calculated with parameter $c$ indicating the extent to which the output is driven by the mean contrast versus the variance in contrast. (G) A power-law nonlinearity ($n$) and a gain ($g$) finally yields the predicted response. Intuitively, the model predicts a large response for increasing contrast and the $c$ parameter determines the extent to which the predicted response is enhanced by variation in contrast across the pRF.

DOI: https://doi.org/10.7554/eLife.47035.014

connections in visual cortex, which preferentially connect cortical columns tuned to the same orientation (*Gilbert and Wiesel, 1983*; *Hata et al., 1988*).

In the OV model, images are first filtered with oriented Gabor patches and combined across quadrature phase (contrast energy), as in the first stage in the SOC model (*Figure 4A*). This results in eight oriented contrast-energy images, one image per orientation band. However, unlike the SOC model, the OV model then sums the contrast energy of each of the orientation images within the pRF, resulting in eight values, one for each orientation band (*Figure 4B* versus Figure 4E). The motivation for this is that gamma oscillations tend to be driven by images dominated by a single orientation; pooling within a band preserves information about the response level of each orientation band summed over space. This is in contrast to the SOC model of BOLD and broadband, which combines the local responses across orientation bands first and then pools over space (*Figure 4E–F*). Finally, the OV model calculates the variance across the eight values, followed by a power-law nonlinearity (exponent $n$) and a multiplication with a gain ($g$) to predict the ECoG response (*Figure 4C*). The key property of the OV model is that it allows variance across orientation to drive the predicted response.

We illustrate the model behavior with a few simple texture patterns (for model parameters $n = 0.5$ and $g = 1$). The OV model predicts a large response when variance across orientations is high and a small response when variance is low. For simplicity, we assume the pRF is as large as the image patch. For a grating pattern, one orientation channel has a large output (high contrast energy), the two neighboring channels have medium outputs, and the five others have small outputs (top left *Figure 5*). When the grating is reduced in contrast (bottom left *Figure 5*), the relative outputs of the orientation bands are unchanged, but the variance is lower. If a second, perpendicular grating is overlaid on the first to create a plaid pattern, matched to the grating in total root mean square (RMS) contrast energy (summed across bands), variance is lower (top right *Figure 5*), and hence the predicted signal is lower. Finally, for a pattern with many orientations at approximately equal contrast, such as in the case of the curved patterns (bottom right *Figure 5*), the variance will be quite low. The OV model thus has two important properties: first, the output increases with contrast within the pRF (high contrast gratings produce higher responses than low contrast gratings), and second, the output increases with increasing variance across orientated contrast energy within

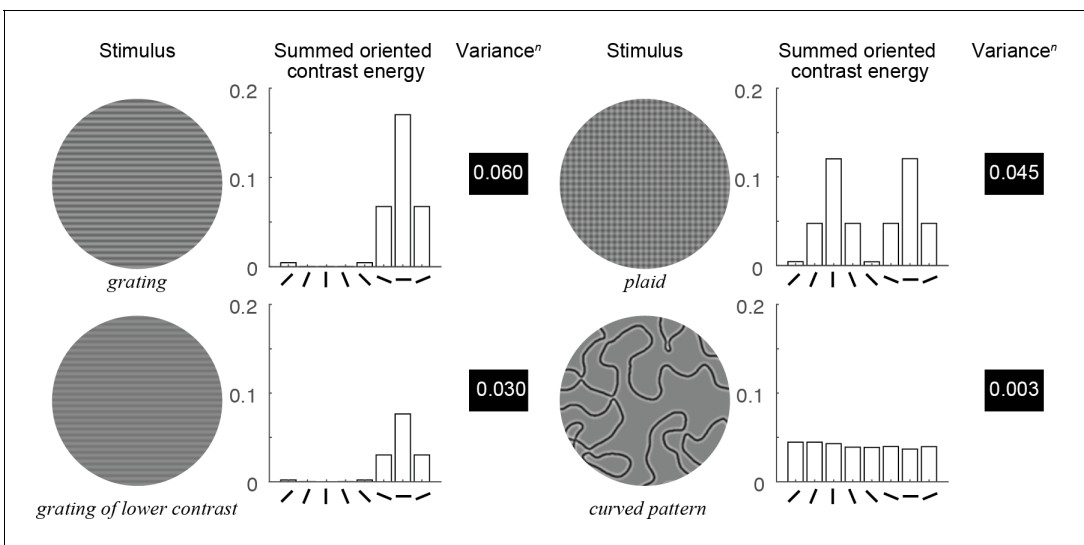

**Figure 5.** Behavior of the OV model. For each example stimulus, the contrast energy within each of 8 orientation bands is summed across the image (bar plots), which simulates the response for a large receptive field spanning the whole image. The variance$^n$ (with $n = 0.5$) across these eight values, monotonically related to the output of the model, is displayed next to the bar plots. This value increases with stimulus contrast (upper left versus lower left) and increases with sparsity of orientations: the high-contrast grating with few orientations present (upper left) has a higher output than the plaid with several orientations present (upper right), which in turn has a higher output than the curved pattern with many orientations (lower right). Code to reproduce this figure can be found on GitHub (*Hermes, 2019*).

DOI: https://doi.org/10.7554/eLife.47035.015

the pRF (gratings produce higher responses than plaids, and plaids produce higher responses than curved patterns, consistent with the observed data in *Figure 2E*, insets).

## Gamma responses are well predicted by a model that is sensitive to variation in orientation content

We tested how well the OV model explains narrowband gamma responses across the 86 images. For all electrodes, gamma responses across the 86 stimuli are largest for the high contrast grating stimuli of different orientations (*Figure 6B*, stimuli 39–46 and *Figure 6—figure supplements 1–*

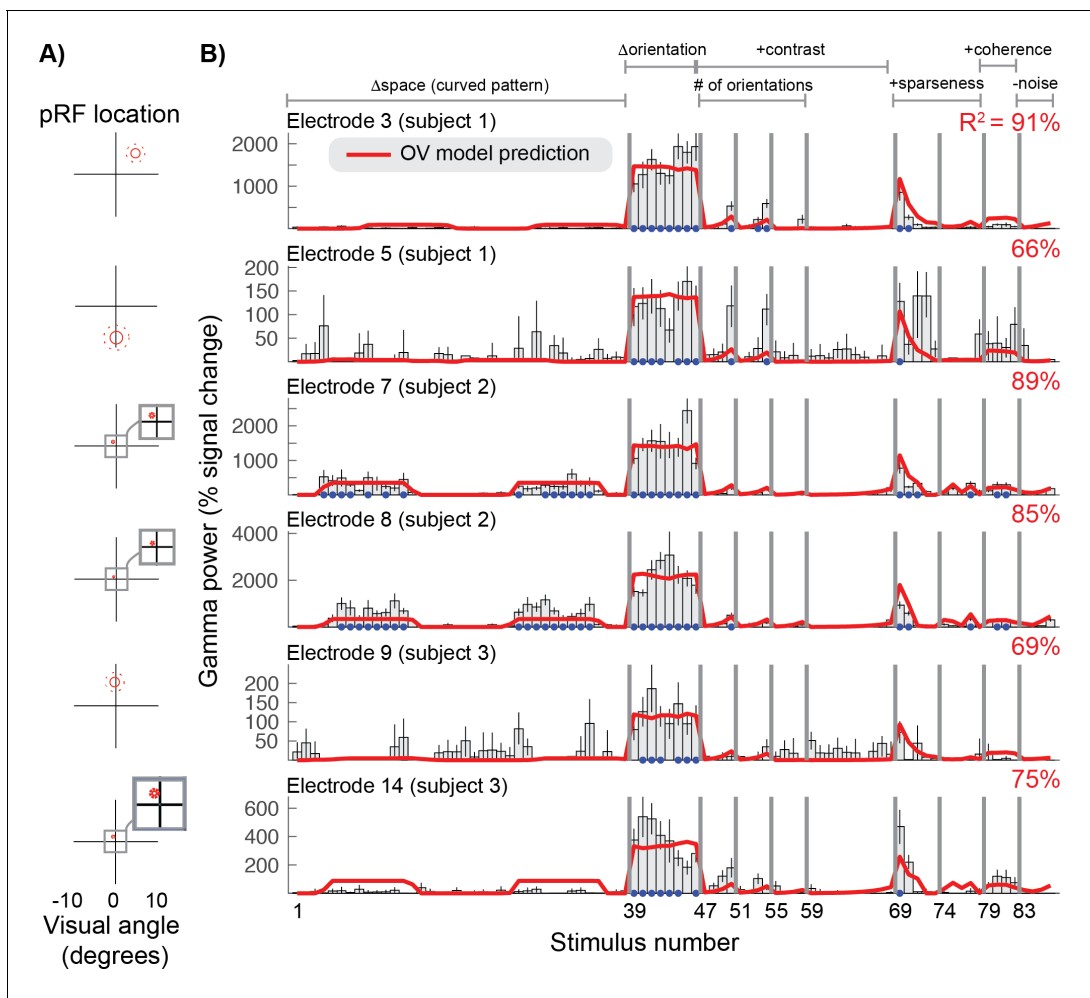

**Figure 6.** Orientation Variance (OV) model predicts selectivity of gamma responses. (A) The population receptive field for each electrode was defined by a Gaussian, indicated by the 1- and 2-sd contours (solid and dotted red lines). (B) The gamma power in percent signal change for six of the 15 electrodes (rows) for all 86 stimuli. Error bars display the 68% range for the bootstrapped responses. The cross-validated predictions of the OV model and overall variance explained ($R^2$) are shown in red. The blue dots indicate that the stimulus response was significantly greater than baseline ($p<0.05$ by bootstrap test). Code to reproduce this figure can be found on GitHub (*Hermes, 2019*).
DOI: https://doi.org/10.7554/eLife.47035.016

The following figure supplements are available for figure 6:

**Figure supplement 1.** Orientation Variance (OV) model predicts selectivity of gamma responses for electrodes 1–8.
DOI: https://doi.org/10.7554/eLife.47035.017

**Figure supplement 2.** Orientation Variance (OV) model predicts selectivity of gamma responses for electrodes 9–15.
DOI: https://doi.org/10.7554/eLife.47035.018

**Figure supplement 3.** Gamma power changes calculated for a timewindow after the evoked response.
DOI: https://doi.org/10.7554/eLife.47035.019

*2*). Gamma is much lower for gratings of reduced contrast (47-50) and plaid patterns (51-55), and is especially weak for circular patterns (stimuli 55–58), with many orientations.

We consider the gamma response to circular patterns to be weak despite the fact that the percent signal change is, for some electrodes, about 20%. In general, the percent signal change is higher for gamma than for broadband, and much higher for broadband than for BOLD (see *Figure 2*). Percent signal change is not easily interpretable without a model of how the signal is generated or a baseline to compare against: for example, a 1% increase in BOLD fMRI signal is likely to correspond to a much larger percent signal change in neuronal firing rate (*Heeger et al., 2000*). Because we do not yet have links between measured gamma, broadband, and BOLD responses in a quantitative model of the neuronal generating signal, we instead compare each type of signal to its own baseline. For the gamma and broadband responses, we compare the stimulus-driven responses against the responses observed during baseline (blank screen). At baseline, the gamma response often reaches 77% signal change (upper boundary of the 68% confidence interval, averaged across electrodes), whereas the broadband response often reaches just 7% signal change (same metric). Hence, a 10% or 20% increase in gamma (for example, observed for the circular gratings) is well within the baseline variability. In contrast, a 10% or 20% increase in broadband is above the baseline variability. Using this approach, we found that the gamma responses to gratings were highly significant compared to baseline (bootstrapping across trials; see Materials and methods), whereas the responses to circular patterns were not for most electrodes (*Figure 6*, all panels).

Overall, the OV model accounted for the pattern of gamma responses well, with an average of 75% cross-validated variance explained across electrodes. The OV model has five parameters: the location and size of the pRF ($x$, $y$ and $\sigma$), a gain factor ($g$) and an exponent ($n$). The pRF center and size were derived from separate data, as they could not be robustly obtained by fitting a model to the gamma responses across the 86 images. This is because the stimuli that varied systematically in spatial position (images 1 to 38) induced very little gamma response. We fixed the $x$ and $y$ position of the population receptive field based on the SOC fits to the broadband data (*Figure 3*). The sigma parameter was derived from the center parameters ($x$, $y$) based on an assumed linear relationship between pRF size and eccentricity as reported in *Kay et al. (2013a)* (*Figure 6A*).

Therefore, the only free parameters in fitting the OV model to the gamma responses were the exponent ($n$) and the gain ($g$). We evaluated a range of values for $n$ ({.1 .2 .3 .4 .5 .6 .7 .8 .9 1}) and directly fit the gain, and used a leave-one-out cross-validation scheme in order to obtain unbiased estimates of model accuracy. Across electrodes an exponent of $n = 0.5$ predicted most variance in the left-out data, and results with this exponent are reported throughout this paper.

The OV model explained significantly more variance in the gamma responses (75%) compared to the SOC model (62%) ($p<0.005$ by a paired t-test on Fisher transform of the $R^2$), see *Figure 7* (red) and *Table 1*. The OV model also explained significantly more variance in the gamma responses compared to a mean model, which predicts the same response level to all stimuli (26%, $p<0.001$, see Materials and methods).

Conversely, the SOC model explained significantly more variance in the broadband responses (80%) compared to the OV model (24%) ($p<0.001$), see *Figure 7* (blue) and *Table 1*. The SOC model also explained significantly more variance in the broadband responses compared to a mean model (41%) ($p<0.001$).

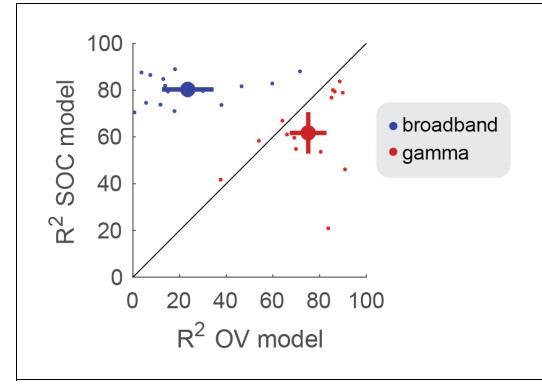

**Figure 7.** OV and SOC model performance on broadband and gamma power. The model performance ($R^2$) was quantified using the coefficient of determination. The x-axis shows the model performance of the OV model fit to the narrowband gamma power (red) and broadband power (blue). The y-axis shows the model performance of the SOC model fit to the gamma power (red) and broadband power (blue). The small dots show the performance for individual electrodes and the large dot indicates the mean + /- two standard errors. Code to reproduce this figure can be found on GitHub (*Hermes, 2019*).
DOI: https://doi.org/10.7554/eLife.47035.021

**Table 1.** Variance explained by different models (coefficient of determination).

| | Model | | |
| --- | --- | --- | --- |
| Data | Mean | SOC | OV |
| Broadband | 41% | 80% | 24% |
| Gamma | 26% | 62% | 75% |

DOI: https://doi.org/10.7554/eLife.47035.020

The model comparisons we have performed suggest that fundamentally different computations underlie broadband and gamma responses. This modeling work complements the experimental observation that the pattern of response is strikingly different for broadband and gamma (*Figure 2*).

## Grating-like features in the pRF strongly drive gamma oscillations

The OV model is sensitive to the distribution of contrast energy across orientations within the pRF, but not to image structure remote from the pRF. Because different electrodes have different pRFs, the output of the model can differ between electrodes in response to the same image, and between exemplars of images taken from the same stimulus class (e.g. natural images) for the same electrode.

To illustrate the importance of taking into account the specific pRF location associated with an electrode, we consider several examples. For a stimulus whose orientation changes over space (*Figure 8A*, left), a large pRF is likely to sample a wide range of orientations. As a result, the spatially summed outputs of different orientation bands are similar, and the variation across these outputs is low. For this reason, for an electrode with a large pRF, the predicted response to the slowly curving pattern (stimulus 10) is smaller than the predicted response to a grating (stimulus 50) (*Figure 8B* left). This prediction is borne out by the data: gamma is much smaller for the curved patterns compared to the gratings for this electrode (*Figure 8B*, left panel). In contrast, a very small pRF, such as in foveal areas of visual cortex (electrode 8), is likely to be exposed to a single dominant orientation, even when the full image contains many orientations. As a result, for a small pRF, the OV prediction is similar for the curved patterns and a grating stimulus (*Figure 8B*, right panel). This example illustrates that it is critical to consider the precise receptive field location of an electrode when investigating gamma oscillations.

The importance of precise pRF locations can also be appreciated by considering responses of a foveal electrode to a variety of curved patterns (*Figure 8C–D*). When the stimulus is relatively sparse, the small pRF may be exposed to a single dominant orientation (stimuli 10 and 77), resulting in large predicted responses, or no contrast at all (stimuli 76 and 78), resulting in little response. When the stimulus is very dense (stimulus 74), the pRF is likely to be exposed to many orientations, resulting in relatively weak responses. This general pattern of responses is observed in the data, demonstrating that it is critical to take into account the specific pRF location for a visual electrode in order to understand the nature of gamma oscillations.

## Predicting gamma oscillations in the context of natural vision

To better understand gamma oscillations in the context of natural vision, we computed the outputs of the OV and SOC models to a large collection of natural images (*Olmos and Kingdom, 2004*). These computations reveal a few interesting patterns. First, the outputs of the two models show some degree of positive correlation, consistent with the fact that both model outputs increase with stimulus contrast within the pRF (*Figure 9*). Second, the responses to gratings are distinct from the responses to natural images, especially for models with larger pRFs (upper panel). This is because the OV output to gratings is unusually high compared to natural images, whereas the SOC output is within or close to the range of outputs to natural images. These patterns are generally consistent across models fit to the 15 electrodes we tested (*Figure 9—figure supplement 1*). High-contrast gratings are often used in studies of gamma oscillations, for example (*Eckhorn et al., 1988*; *Hoogenboom et al., 2006*; *Rohenkohl et al., 2018*); using these stimuli as a benchmark, we find that the OV outputs for natural images are, in general, strikingly low. However, natural images occasionally have oriented high-contrast energy within the pRF and these give rise to large OV output (*Figure 9*). Hence, the image properties to which the OV model is sensitive exist in natural images,

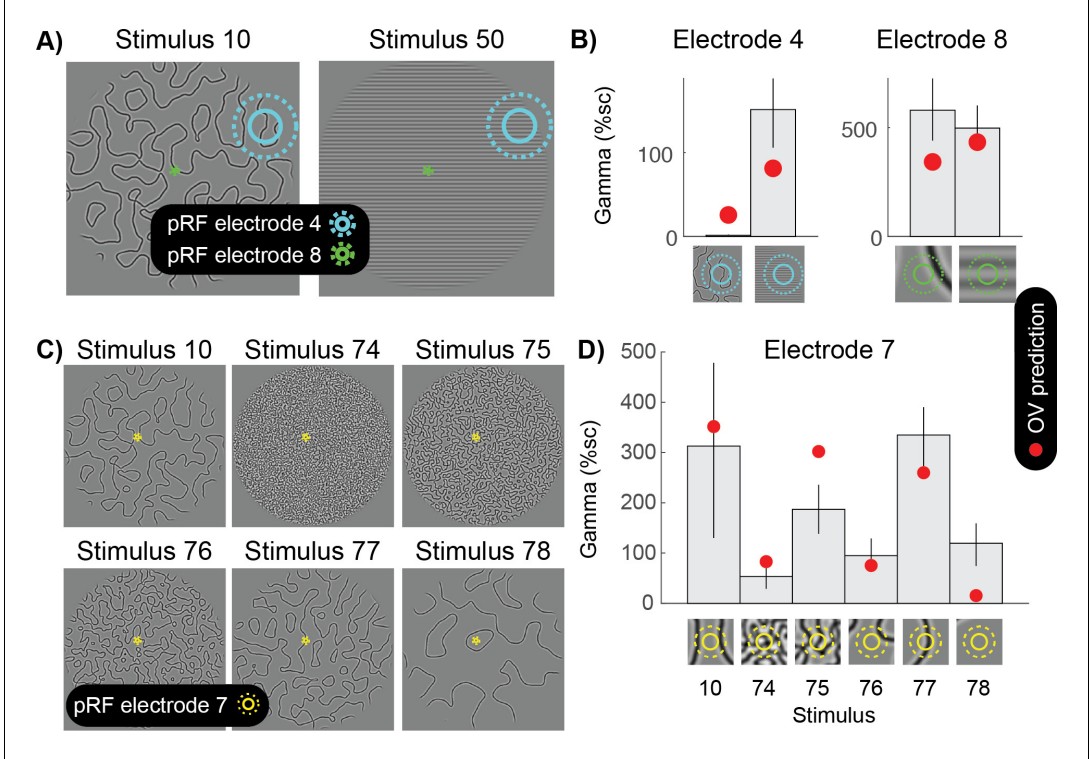

**Figure 8.** Grating-like features in the pRF strongly drive gamma oscillations. (**A**) The population receptive fields (pRF) of two electrodes are overlaid on two different curved patterns (stimulus 10 and 50). Electrode four has a large pRF, while electrode eight has a very small pRF. (**B**) The left panel shows that for electrode 4, the OV model predicts a small response for the curved lines and a large gamma response for the grating pattern (red dots). As predicted, there is a small gamma response for the curved lines and a large response for the grating stimulus. Blue circles on the bottom zoom into the stimulus in the pRF. The right panel shows that for electrode 8, the OV model predicts a similar response for the image with curved lines and the image with a grating. As predicted, there is a large gamma response for the curved lines and a large response for the grating stimulus. Green circles on the bottom zoom into the stimulus in the pRF and this shows that from the curved lines, only a relatively straight line falls in the pRF. (**C**) Six different images with curves differing in sparseness (stimulus 10, 74, 75, 75, 76, 77 and 78). The population receptive field (pRF) of electrode seven is overlaid with one and two standard deviations (solid yellow and dotted yellow). (**D**) The OV model predicts the largest response when a grating-like feature hits the pRF (red dots). As predicted, the largest gamma responses are observed when the pRF contains grating-like features. Yellow circles on the bottom zoom into the pRF content of each of the stimuli. Error bars display the 68% confidence interval (across bootstraps), and the close up of image in the pRF show the outline of the pRF at 1 and 2 standard deviations (straight and dashed). Code to reproduce this figure can be found on GitHub (**Hermes, 2019**).

DOI: https://doi.org/10.7554/eLife.47035.022

but in far lower quantities than they do for oriented gratings (and other stimuli such as bars [**Gray and Singer, 1989**; **Gray et al., 1989**] and circular gratings [**Hoogenboom et al., 2006**] that are locally similar to oriented gratings). Note that the range of the OV outputs for these natural images (on the order of ~100%) is comparable to the gamma band responses measured in monkey ECoG to gray-scale natural images reported by **Brunet et al. (2015)**. To the degree that the OV model accurately predicts narrowband gamma responses and the SOC model accurately predicts broadband responses, these results indicate that for most natural gray-scale images (with occasional exceptions), relatively low levels of gamma oscillations are expected.

Finally, we note that the absolute scale of the responses differs substantially between the models, with a larger maximal response for the OV model, especially for gratings. For example, there is up to 2000% signal increase over baseline in the OV model compared to 300% for the SOC model (**Figure 9B**). This difference in amplitude exists not only in the model outputs, but also in the data, as shown earlier (e.g., **Figures 2**, **3** and **6**).

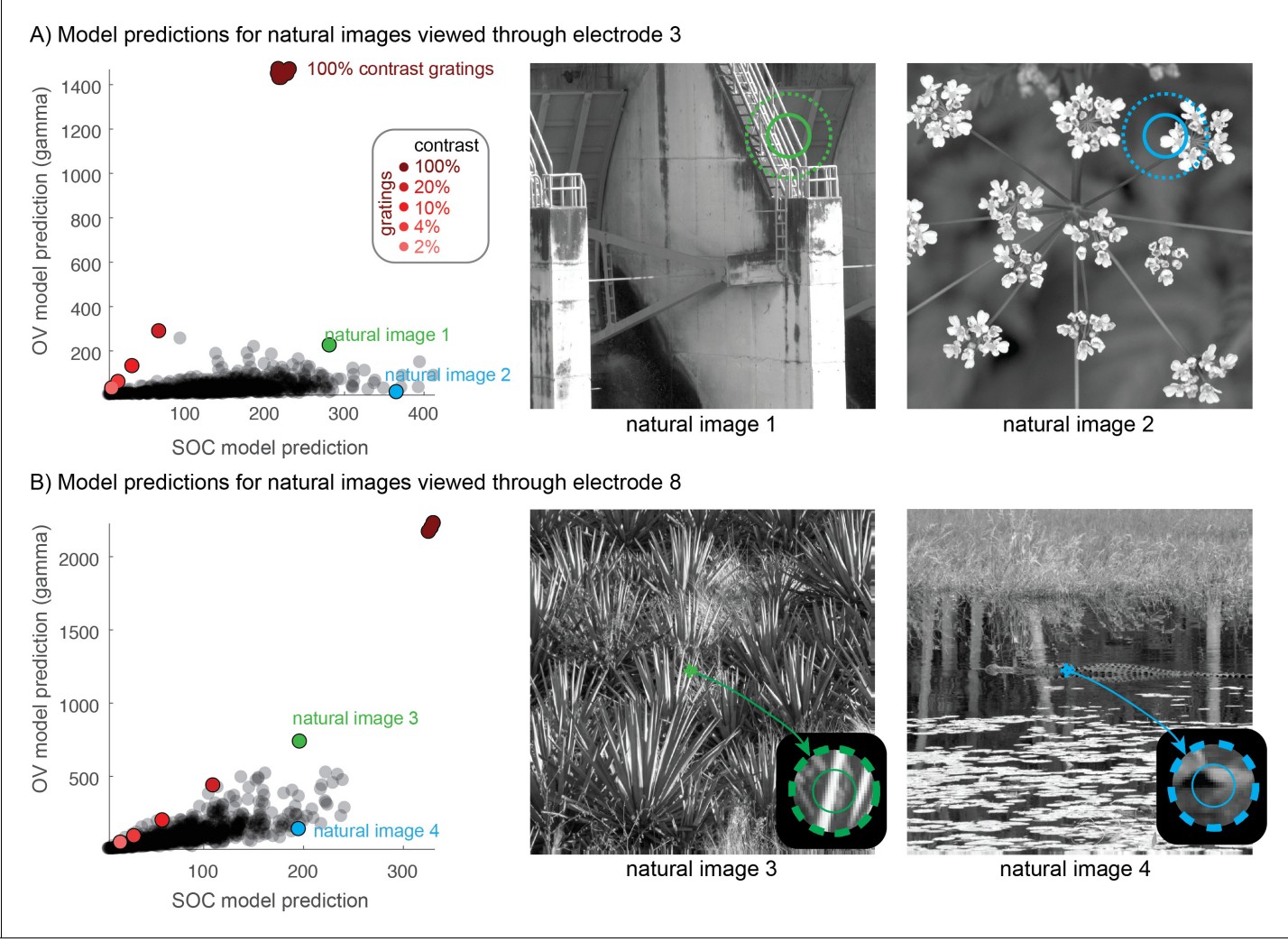

**Figure 9.** OV and SOC model predictions for images of natural scenes. (**A**) The OV and SOC outputs are plotted for a set of gray-scale photographs of scenes, with model parameters from electrode 3. The units are in percent signal change, as in *Figures 3* and *6*. Each gray dot is the output of the two models for one image. The red dots are the model outputs for grating stimuli of varying contrast. The cluster of red dots at 100% contrast displays high-contrast gratings of different orientations (stimuli 39–46). The green and blue dots correspond to two images with large OV and SOC outputs, respectively. The right panels show these two images with the electrode pRF location superimposed (1 and 2 SDs). Natural image 1, with a high OV output, has image features in the pRF that look like a grating. The OV output to images of natural scenes are much lower than the responses to high contrast gratings. (**B**) Same as panel A, but for electrode 8, including a zoom into the pRF location. Code to reproduce this figure can be found on GitHub (*Hermes, 2019*).

DOI: https://doi.org/10.7554/eLife.47035.023

The following figure supplement is available for figure 9:

**Figure supplement 1.** OV and SOC model predictions for images of natural scenes for all electrodes.

DOI: https://doi.org/10.7554/eLife.47035.024

## Discussion

Narrowband gamma power is highly stimulus dependent, as observed in a number of studies. For example, there are large gamma oscillations in response to bars and gratings measured by electro-encephalography (EEG) (*Murty et al., 2018*; *Scheeringa et al., 2011*), magnetoencephalography (MEG) (*Hoogenboom et al., 2006*; *Muthukumaraswamy and Singh, 2009*; *Muthukumaraswamy and Singh, 2008*), and microelectrode local field potentials (LFP) (*Gray and Singer, 1989*; *Gray et al., 1989*). In contrast, other studies have found little gamma response to noise patterns (*Bartoli et al., 2019*; *Hermes et al., 2015*; *Jia et al., 2013*) and many natural images

(*Hermes et al., 2015*; *Kayser et al., 2003*). Here, we recorded ECoG data from three subjects viewing 86 static, band-passed, grayscale images and developed an image-computable model to predict the level of gamma oscillations observed for these 86 stimuli.

## Stimulus selectivity of gamma oscillations in visual cortex

The most salient observation is that the narrowband gamma responses were much sparser than the broadband responses. The gamma responses were large for only a few stimuli tested (high contrast gratings), and relatively small or even within the experimental noise for most other stimuli (e.g., noise patterns and textures with a lot of curvature). The sparseness in the narrowband gamma responses was not due to a lack of image contrast or measurement sensitivity. Many of the stimuli that elicited little measured narrowband gamma responses nonetheless elicited large broadband responses. In fact, large broadband responses were observed for most stimuli tested, including all stimuli with high contrast within the electrode's receptive field (pRF). The much greater sparseness in the narrowband gamma compared to broadband responses was a highly consistent finding, observed in every electrode studied (n = 15, coming from three subjects and spanning V1-V3). Thus, despite the modest number of subjects tested, it is highly likely that these findings will generalize and replicate.

The sparseness in the narrowband gamma responses is striking not just in comparison to the broadband responses reported here, but also in comparison to fMRI responses to the same stimuli measured previously (*Kay et al., 2013b*). Like broadband, the fMRI response was large for all stimuli with high contrast in the pRF and tended to be larger for curved patterns than for gratings. The similarity in stimulus selectivity between fMRI and broadband is consistent with a number of other studies indicating good agreement between the two types of signals. These studies span multiple cortical areas and stimulus manipulations, including motor cortex and finger movements (*Hermes et al., 2012*), auditory cortex and natural movies (*Mukamel, 2005*), ventral temporal cortex and category selectivity (*Jacques et al., 2016*), occipital cortex and spatial summation (*Winawer et al., 2013*), and occipital cortex and pattern selectivity (*Hermes et al., 2017b*).

While the similarity between the broadband and fMRI responses appears to be widespread across the brain, our conclusions about the sparseness of the narrowband gamma responses are specific to visual cortex. The circuitry in primary visual cortex is unique, with its retinotopic structure, ocular dominance stripes, and orientation columns (*Wandell, 1995*). The fact that two different brain regions can exhibit oscillations at similar frequencies does not mean that their function is shared or that the underlying neurophysiological mechanisms generating the oscillations are the same. For example, gamma oscillations in the tectum of the barn owl (*Sridharan et al., 2011*), the nucleus accumbens in humans (*Miller et al., 2019*) or the hippocampus of the rodent (*Buzsáki et al., 1983*) likely have different circuit properties, neuronal origins and computational functions compared to gamma oscillations in primate V1. In fact, even within the same cortical region—mouse V1—there appear to be distinct types of gamma rhythms with different stimulus sensitivities and different biological origin (*Saleem et al., 2017*). Conclusions about gamma oscillations in human visual cortex can thus not be directly transferred to other systems without careful testing.

## The orientation-variance (OV) model of gamma responses

Because of the similarity in stimulus selectivity between broadband and fMRI, the same model (SOC) was appropriate to explain both measurements (*Kay et al., 2013b*). The strikingly different stimulus selectivity of the narrowband gamma responses motivated us to develop a novel image-computable model, the Orientation-Variance (OV) model. The OV model is sensitive to the variation across the spatially pooled outputs of the orientation channels in the population receptive field. The two models share a common first stage, in which contrast energy is computed. As a result, both models predict that response amplitudes should increase with stimulus contrast, in agreement with the observation that fMRI, broadband, and narrowband ECoG responses all increase with contrast. The subsequent calculations of the two models differ, particularly in the order of operations. The SOC model first sums across orientation at each spatial location and then computes variance across space. As it sums across orientations, it loses sensitivity to variance across orientations. The OV model does the opposite, first summing across space (within each orientation band) and then computing variance across orientations. As it sums across space, it loses sensitivity to variance across

space. The difference in the order of operations has a large effect on the image properties that are emphasized by the model. The SOC model is blind to the variance in orientations within the population receptive field and the OV model is blind to the variance in contrast energy across space within the population receptive field. Also, the SOC model penalizes images (lowers the response) when contrast is high everywhere, akin to surround suppression. Because the OV model first sums across space, responses grow with stimulus size, as is observed for gamma oscillations but not spiking (*Gieselmann and Thiele, 2008*; *Jia et al., 2013*; *Ray and Maunsell, 2011*; *Self et al., 2016*) or BOLD (*Press et al., 2001*; *Zenger-Landolt and Heeger, 2003*; *Zuiderbaan et al., 2012*).

For both models, the computations are made within the population receptive field. Therefore, the question of whether gamma is induced by an image, like the question of whether any neural signal is induced by an image, is only sensible if we take into account the *specific receptive field location* of the neurons or neuronal population under consideration, as shown in *Figure 8*. The OV model implies that the number of orientations within the population receptive field needs to be taken into account in order to predict gamma oscillations.

In this study we used carefully designed stimuli to understand which image features drive gamma oscillations. There are several other parameters that could be explored. For example, color has recently been shown to strongly modulate gamma (*Peter et al., 2019*; *Shirhatti and Ray, 2018*). In addition, other features present in natural images were not explored in our measurements, but only with simulations. Using well controlled stimuli, our data show that image contrast (first order contrast) and variations in contrast (second order contrast) are, in and of themselves, not sufficient to elicit robust gamma oscillations.

While the OV model was in development, another group independently tested models of gamma oscillations in visual cortex using images of complex objects and scenes, reaching opposite conclusions (*Brunet and Fries, 2019*). Inferring the stimulus properties that drive neural responses from such studies is difficult because of unmodeled correlations that are widespread in natural images (*Rust and Movshon, 2005*). The advantage of carefully controlled stimulus properties is that the interpretation of the results is more straightforward, as is the link between the computations in the model and the pattern of responses. Future work will be required to link our models of spatial pattern to models developed for natural images and models of chromatic sensitivity.

## Different origins of broadband and narrowband gamma

The different stimulus selectivity in the broadband and the gamma responses was emphasized by choices we made both in the selection of stimuli and in the analysis of the data. For example, had we only used gratings of varying contrast, the gamma, broadband, and BOLD signals would have been found to collectively rise together (though perhaps differing in the precise shape of the contrast response functions [*Henrie and Shapley, 2005*; *Lima et al., 2014*]). Similarly, when using only grating stimuli which elicit large gamma responses, for example (*Scheeringa et al., 2011*), trial-to-trial variance in responses may reflect global factors such as attention or arousal; these global factors might modulate multiple signals, such as gamma power and BOLD, thereby causing the two signals to be correlated within the experimental paradigm. By systematically exploring variation in stimulus properties such as the number of component orientations (gratings vs plaids vs circular patterns), we were able to reveal opposing effects on broadband and gamma responses, similar to the effects of manipulating stimulus size, which has opposite effects on the level of broadband and gamma responses (*Ray and Maunsell, 2011*). These different response patterns show the importance of testing a wide range of stimuli.

The method of separating the ECoG measurement into two components was also important. Had we simply characterized the gamma response as the band-limited power increase over baseline, the gamma responses would have appeared less sparse. For example, the noise patterns in *Figure 1* cause power increases in the gamma band (~30–80 Hz) for two of the three electrodes. However, because the power increase spans higher frequencies and contains no clear peak, the response is better characterized as a broadband response, rather than as a narrowband gamma oscillation (*Lopes da Silva, 2013*). This is reflected in how we compute the two signals. We do not assume a direct relationship to temporal frequency bands since the two signals can overlap in their spectra; rather, we take a model-based approach in which we separate out a peaked response (oscillatory) from one that is non-peaked (broadband). This approach is motivated by the fact that the two responses can be modulated independently (e.g., the peaked response decreases with the number

of orientations, whereas the broadband response increases). Additionally, independent modulation of a broadband component spanning frequencies from 1 to 200 Hz has been previously demonstrated by principal components analysis of spectral power (*Miller et al., 2009a*). Studies that do not explicitly separate broadband and narrowband gamma oscillations might draw conclusions about the function of narrowband oscillations when the signals measured could, in fact, reflect broadband responses.

We observed that percent signal change in gamma responses is much larger than that for broadband responses. Broadband responses are thought to arise from asynchronous neural activity (*Miller et al., 2009b*), which results in substantial cancellation in the pooled field potential (*Butler et al., 2017*; *Hermes et al., 2017b*; *Krusienski et al., 2011*; *Winawer et al., 2013*). Narrowband gamma responses, in contrast, are thought to reflect synchronous activity across a neuronal population (*Hasenstaub et al., 2005*; *Jia et al., 2013*). The synchronous response, even if it comes from a much smaller neuronal population, can result in a much larger macroscopic field potential (*Butler et al., 2017*; *Hermes et al., 2017b*; *Winawer et al., 2013*). This likely explains why gamma oscillations, even when the percentage signal change is quite large, can show little correlation with the BOLD signal (*Butler et al., 2017*; *Hermes et al., 2017b*) or multiunit activity (*Jia et al., 2013*; *Ray and Maunsell, 2011*). Hence the large size of the field potential generated by gamma oscillations does not imply a high level of neuronal population activity in terms of energy demand or spike rates. Similarly, a large neuronal response can occur in the absence of gamma oscillations. For example, multiunit recordings in macaque V1 and MT to sweeping bars showed oscillating responses in only 2 of 424 recordings, while noting increases in firing rates and broadband LFP (*Young et al., 1992*). In contrast, substantial increases in broadband power generally correlate with both high energy consumption (*Hermes et al., 2017b*; *Winawer et al., 2013*) and high firing rates (*Manning et al., 2009*; *Ojemann et al., 2013*; *Ray and Maunsell, 2011*) (though for an exception see [*Leszczynski et al., 2019*]).

## Relevance for neuronal and cognitive function

The highly specific stimulus selectivity of narrowband gamma raises questions about the potential functions of this signal. A wide range of cognitive and neural functions have been attributed to gamma oscillations, including perceptual binding of visual features (*Eckhorn et al., 1988*; *Gray et al., 1989*), prioritizing communication of certain visual information over other information (*Fries, 2005*), and visual awareness (*Engel and Singer, 2001*). These theories do not lead to quantitative predictions about the amplitude of gamma oscillations that one would expect from specific images. However, it is reasonable to expect that if gamma oscillations are a leading cause of visual awareness, or of feedforward visual communication, then the oscillations should be observed for any clearly visible stimulus. If there is a general role of gamma oscillations in typical brain function, it is not clear why oscillations would be large for oriented gratings and minimal or possibly absent for curved patterns or stimuli with multiple orientations. At high contrast, all of these stimuli are easily visible and all of them elicit robust signals in visual cortex as measured with fMRI and broadband ECoG. Moreover, these stimuli would be expected to elicit strong multiunit responses, given typical models of action potentials in neuronal populations in primary visual cortex (e.g., [*Carandini et al., 2005*; *Rust et al., 2005*]).

Finally, it is reasonable to suppose that stimuli require transmission to downstream visual areas in support of recognition and behavior. In fact, fMRI studies show that stimuli such as circular patterns, which elicit no detectable narrowband gamma signal beyond baseline signal levels, do evoke large and reliable BOLD signals in higher visual areas such as hV4 (*Kay et al., 2013b*). Hence visual signals are transmitted along the visual hierarchy for stimulus features to which we measure no reliable narrowband gamma signals in the corresponding retinotopic locations of V1. This suggests that it is unlikely for gamma oscillations to be the primary means of communicating long-range feedforward visual information. Nonetheless, there are several reasons why we cannot entirely rule out a possible role of narrowband visual gamma oscillations in various cognitive tasks. One is that theories proposing a role for gamma oscillations are pitched at a general level and do not make quantitative predictions about the amplitude of the oscillations for specific stimuli or conditions. Thus, it is difficult to falsify such theories. Secondly, we acknowledge that there may be small oscillations that are not detectable using our current measurement methods. It is therefore possible that very small oscillations exist and serve critical functions. In this study, rather than asking *whether* a signal such as

gamma oscillations is present or absent (in one or a few stimulus conditions), we sought to develop a predictive, quantitative model that provides a more detailed characterization of the signal (across many stimulus conditions). We believe that this has provided important insights into the nature of gamma oscillations.

The OV model developed in this study is agnostic with respect to the downstream effects that gamma oscillations cause when they are present. Large oscillations are likely to influence spiking excitability (*Jensen et al., 2007*) or information transfer (*Besserve et al., 2015*). A complete theory of the relationship between gamma oscillations and any cognitive or neural function, however, must also account for cases where oscillations are small or absent, and should also be able to predict their level given the visual input.

## Gamma oscillations and gain control

Although our study was not designed to test a particular function of gamma oscillations, our results, as well as previous reports, provide some important clues. Both in this report and elsewhere (*Gieselmann and Thiele, 2008*; *Henrie and Shapley, 2005*; *Jia et al., 2013*; *Ray and Maunsell, 2011*), large-amplitude gamma oscillations are found for stimuli that are high in contrast, spatially extended, and with few orientations. These three image properties—contrast, spatial extent, and limited orientations—all produce larger outputs in the OV model, and interestingly, are all associated with gain control or suppression in neuronal circuits.

First, stimulus contrast has been linked to inhibition in divisive normalization models of primary visual cortex (*Heeger, 1992*). According to this model, gain control increases with local stimulus contrast, possibly via shunting inhibition (an increase in membrane conductance) or a reduction in recurrent amplification (*Sato et al., 2016*). Although neuronal responses such as spike rates tend to increase with contrast, the rate of increase is slower at higher contrast (*Albrecht and Hamilton, 1982*), consistent with mechanisms of increasing gain control at higher contrast (*Albrecht and Geisler, 1991*; *Heeger, 1992*). Second, stimulus extent is linked to suppression in that larger stimuli stimulate the inhibitory surrounds of neuronal receptive fields, thereby reducing the neuronal response (*Allman et al., 1985*). Third, for a large stimulus, surround suppression is more effective when an annulus and a central stimulus match in orientation (*Cavanaugh et al., 2002*; *DeAngelis et al., 1994*; *Knierim and van Essen, 1992*).

In summary, each of these three stimulus properties (high contrast, large spatial extent, limited orientations) is associated with more gain control or suppression as well as a larger OV output, consistent with the interpretation that gamma oscillations are a biomarker of gain control or normalization, an idea previously proposed based on physiology data from macaque visual cortex (*Ray et al., 2013*). Ray et al., (2013) showed that stimulus manipulations thought to increase normalization lead to larger amplitude gamma oscillations. For example, when a null-motion stimulus is added to a preferred motion stimulus, spike rates decrease, indicating an inhibitory effect of the null motion stimulus, and gamma oscillations increase in amplitude (*Ray et al., 2013*). Similar to our study, this shows that a stimulus configuration that increases inhibition also increases the amplitude of gamma oscillations. We note that the link between gamma oscillations and gain control does not necessarily indicate what causal role, if any, the oscillations have in neural processing. At a minimum, the oscillations may serve as a biomarker of gain control circuits, useful to the experimenter but not necessarily to the organism producing them. Whether or not the oscillations are critical for implementing gain control requires further study.

One exception to the link between inhibition and gamma power in this study is the effect of the number of component orientations: increasing the number of superimposed gratings decreases the OV output and the power of narrowband gamma (*Figure 3*), yet causes an increase in cross-orientation suppression in visual cortex (*Bonds, 1989*; *Morrone et al., 1982*). This breaks the pattern by which stimulus properties that cause more inhibition also cause larger-amplitude gamma oscillations. Cross-orientation suppression and surround suppression differ in several ways, including in their temporal properties, with surround suppression slightly delayed (*Smith et al., 2006*). This supports the possible interpretation that cross-orientation suppression is inherited from earlier processing in a feedforward manner, whereas surround suppression depends on intra-cortical connections (either within a cortical area and/or via feedback). A large increase in gamma oscillations may therefore reflect locally implemented inhibition, as in surround suppression, but not inherited suppression, as in cross-orientation suppression. More generally, the fact that these two types of suppression likely

have different underlying mechanisms, and different effects on gamma oscillations, highlights the importance of considering the circuit-level implementation of computations such as suppression and gain control.

At a more abstract level, the same image properties that are associated with gain control can also be described as signatures of image redundancy (or predictability). Since gamma oscillations tend to increase in the presence of gain control, they can also be described as increasing in the presence of image redundancy or predictability (*Vinck and Bosman, 2016*). In fact, early descriptions of center-surround visual receptive fields proposed that surround suppression was part of a coding strategy to compress signals elicited by natural images which contain a lot of redundancy in the form of spatial correlations (*Barlow, 1961*).

## Gamma oscillations and neuronal circuits

While there is growing evidence that gamma oscillations increase in the presence of gain control or inhibition, the circuitry underlying these inhibitory mechanisms are not firmly established. For example, contrast gain control modeled as divisive normalization might be implemented biologically as shunting inhibition (synaptic inhibition which changes the neuronal membrane conductance) (*Carandini and Heeger, 1994*; *Carandini et al., 1997*). In this implementation, the notion of inhibition in the model (signal reduction by division) is literally an increase in inhibitory neural signals.

Alternatively, normalization could be implemented by a circuit that reduces excitation, which in turn also *reduces* inhibition rather than increases it, as is the case for inhibition-stabilized networks (*Ozeki et al., 2009*; *Tsodyks et al., 1997*). In inhibition-stabilized networks, the un-normalized state (e.g., low contrast, no suppressive surround) has a high level of recurrent excitation and inhibition, with the inhibition serving to stabilize the network. Stimulus manipulations that result in an increase in gain control, such as the addition of a surrounding stimulus or an increase in contrast, paradoxically result in a withdrawal of inhibition. This, in turn, destabilizes the network, allowing the activity to either die off or explode. Gamma oscillations may be more likely to arise (or to increase in amplitude) in this destabilized state. In the extreme, large, high-contrast oriented gratings can even trigger seizures in patients with photosensitive epilepsy or cause discomfort in healthy subjects (*Harding et al., 2005*; *Hermes et al., 2017a*; *Wilkins et al., 1984*).

A large number of studies have tried to explain the circuit mechanisms that underlie gamma oscillations, including via explicit computational models (*Ainsworth et al., 2012*; *Buzsáki and Wang, 2012*; *Womelsdorf et al., 2014*). Some of these models are formulated with the goal of explaining a particular stimulus sensitivity of computational function, and thus predict modulations in the amplitude of gamma oscillations as a function of labeled stimulus properties, such as size or contrast (*Jia et al., 2013*). This particular model consists of an excitatory and inhibitory neuron, representing a cortical hypercolumn. Each neuron is driven by independent inputs and projects to itself and the other neurons. In addition, a global component is included that represents more distributed inputs, such as horizontal connections or input from higher order visual areas. The global component receives input from the excitatory neurons and sends output to the inhibitory and excitatory neurons. This model explains why firing rates and gamma frequency and power can change with different types of inputs. Relating this model to our data, we can potentially imagine the global component being driven by long-range horizontal connections between orientation columns in V1 with the same orientation preference (*Angelucci and Bressloff, 2006*). Yet to our knowledge, none of the circuit models for gamma oscillations operates on visual inputs in the sense of arbitrary pixel intensities, as does our image-computable OV model and various models tested recently by *Brunet and Fries (2019)*. Our model, however, (as well as those from Brunet and Fries) does not describe the neuronal circuitry that produces the oscillations. A more complete understanding of this intensely studied neural signal will likely require a unified account that both generalizes to arbitrary images and also specifies the circuitry that underlies oscillations.

## Conclusions

Gamma oscillations in human visual cortex are elicited by distinct types of visual inputs that differ from fMRI BOLD and ECoG broadband responses. We developed an image-computable 'orientation-variance' model, which accounts for the amplitude of gamma oscillations across many stimuli. In this model, gamma oscillations are driven by increases in contrast and by variance across orientation

channels in the population receptive field. These findings are consistent with the proposal that gamma oscillations reflect circumstances in which neural circuits exhibit strong normalization or gain control.

## Materials and methods

### Ethics statement and subjects

ECoG data were recorded in three subjects (mean age 28, two women) who had electrodes implanted for the clinical purpose of epilepsy monitoring. Subjects gave informed consent and the study was approved by the Stanford University IRB and the ethics committee at the University Medical Center Utrecht in accordance with the 2013 provisions of the Declaration of Helsinki.

### Stimuli and task

Static visual images were viewed from a distance of ~50 cm and spanned approximately 20 degrees of visual angle. Images were presented for 500 ms (stimulus period), followed by a gray screen for 500 ms (baseline period). There were 86 different gray scale images that contain spatial frequencies of 3 cycles per degree. Images included curved lines with varying apertures spanning parts of the visual field, full screen gratings varying in orientation, plaids, circular patterns with 16 orientations and curved lines varying in contrast, gratings and curved lines varying in sparseness, gratings varying in coherence and curved lines with different levels of noise. These stimuli could roughly be grouped into the categories of SPACE (1 to 38), ORIENTATION (39 to 46), CONTRAST (47 to 68), SPARSITY (69 to 78), and COHERENCE (79 to 86) All images are shown in *Figure 2—figure supplement 1–5*. Images were created in similar manner as in *Kay et al. (2013b)*, with only images 79-86 being a new category. Each image was repeated several times (subject 1: 15 times, subject 2: nine times, subject 3: 12 times).

### ECoG procedure

ECoG electrodes were placed on the left hemisphere in subject one and on the right hemisphere in subjects 2 and 3. ECoG data were recorded at 1528 Hz through a 128-channel Tucker Davis Technologies recording system (http://www.tdt.com) (subjects 1 and 2) and at 2048 Hz through a 128-channel Micromed recording system (subject 3). To localize electrodes, a computed tomography (CT) scan was acquired after electrode implantation and co-registered with a preoperative structural MRI scan. Electrodes were localized from the CT scan and co-registered to the MRI, and positions were corrected for the post-implantation brain shift (*Hermes et al., 2010*). Electrodes that showed large artifacts or showed epileptic activity, as determined by the patient's neurologist were excluded, resulting in 116/107/54 electrodes with a clean signal. Offline, data were re-referenced to the common average, low pass filtered and the 1528 Hz data were resampled at 1000 Hz for computational purposes using the Matlab resample function. Line noise was removed at 60, 120 and 180 Hz (Stanford) using a third order Butterworth filter, data from UMC Utrecht did not contain much line noise and were not filtered. In further analyses, we only included electrodes that were located on visual areas V1, V2 or V3 with population receptive fields that were within the stimulus (~10 degrees from the fovea).

### ECoG analyses

#### Time frequency analysis

Time frequency analysis was performed around stimulus onset (−500 to 1000 ms) with a multitaper approach (*Percival and Walden, 1993*) using chronux (http://www.chronux.org/; *Mitra and Bokil, 2008*). A moving window of 200 ms (with overlap of 50 ms) and the use of 5 tapers result in a frequency resolution of 5 Hz, with a spectral smoothing of ±15 Hz. To normalize the responses to baseline, the average spectrum from all inter trial intervals 250–500 ms after stimulus offset was computed and divided from every time bin. The base 10 log was then computed on this normalized power and plotted (*Figure 1*).

## Spectral analysis

We calculated power spectra and separated ECoG responses into broadband and narrowband gamma spectral power increases in similar manner as before (*Hermes et al., 2015*). For each stimulus and baseline epoch, the average power spectral density was calculated every 1 Hz by Welch's method (*Welch, 1967*) with a 200 ms window (0–500 ms after stimulus onset, 100 ms overlap). A Hann window was used to attenuate edge effects. The evoked potential did not have a large effect on our results: phase locking plots of the data indicated that most power resided below 30 Hz (*Figure 1—figure supplement 1*) and similar results were obtained when analyzing the window from 200 to 500 ms (after the evoked potential is complete, *Figure 3—figure supplement 3*, *Figure 6—figure supplement 3*).

ECoG data are known to obey a power law and to capture broadband and narrowband gamma increases separately, the following function (*F*) was fitted to the average log spectrum from 30 to 200 Hz (leaving out 60 Hz line noise and harmonics) from each stimulus condition:

$$F(x) = (\beta_{broadband} - nx) + \beta_{narrowband}G(x|\mu,\sigma)$$

In which,

$$x = log_{10}(frequency)$$

$$G(x|\mu,\sigma) = \frac{1}{\sigma\sqrt{2\pi}}e^{-\frac{(x-\mu)^2}{2\sigma^2}}$$

with $0.03 < \sigma < 0.08$ and $30 Hz < 10^\mu < 80 Hz$.

This function allows broadband and gamma components to vary independently in the measured data. The slope of the log-log spectral power function (*n*) was fixed for each electrode by fitting it based on the average power spectrum of the baseline.

## Bootstrapping and confidence intervals

Confidence intervals were estimated using a bootstrap procedure: for each stimulus condition C with $N_c$ trials, $N_c$ trials were drawn randomly with replacement and power spectra were averaged. The function *F* was fit to the average $log_{10}$ power spectrum from these trials and the β parameters were estimated. This was repeated 1000 times, resulting in a distribution of broadband and narrowband weights. The same was done to calculate confidence intervals for the baseline period.

## Converting model estimates to percent signal change

The β parameters in function *F(x)* have units of $log_{10}$ power and from this we derived the percent signal change in broadband and gamma power. The percent signal change is defined as:

$$percent\ signal\ change = \left(\frac{taskPower}{baselinePower} - 1\right) * 100$$

in which,

$$\frac{broadbandTaskPower}{broadbandBaselinePower} = 10^{broadband}$$

with

$$broadband = \beta_{broadbandTask} - \beta_{broadbandBaseline}$$

and,

$$\frac{gammaTaskPower}{gammaBaselinePower} = 10^{gamma}$$

with

$$gamma = \beta_{gammaTask}$$

Given the assumption that gamma power is close to zero during the baseline period.

## Significance testing

To test whether responses to a particular stimulus differed significantly from baseline, we performed a bootstrap test. For each electrode, we used the 1000 bootstraps for each stimulus and for the baseline period and did a one-sided test of how many times the difference (bootstrapped stimulus minus baseline response) was larger than zero. We report a significant response for $p < 0.05$.

## Model

### Image preprocessing

Images of $800 \times 800$ pixels were downsampled to $240 \times 240$ pixels for computational purposes and converted into a contrast image: all pixel values between 0 and 254 were rescaled to a range from zero to one and the background luminance was subtracted (0.5), resulting in all pixel values in a range from $-0.5$ to 0.5 with the background corresponding to zero. Images were zero padded with 15 pixels on each side to reduce edge effects, resulting in images of size $270 \times 270$.

### Oriented contrast energy (step 1)

After preprocessing, the images were filtered with isotropic Gabor filters with eight different orientations and two quadrature phases covering positions on a grid of $135 \times 135$. Since the stimuli were band-pass, filters had one spatial scale with a peak spatial frequency of 3 cycles per degree. The filters were scaled such that the response to a full-contrast optimal grating was 1. After quadrature-phase filtering, the outputs were squared, summed, and square-rooted and the results can be expressed as oriented contrast energy (*Figure 4A*):

$$oriented\ contast\ energy_{pos,or} = \sqrt{\sum_{ph}\left(stimulus \cdot filter_{pos,\ or,\ ph}\right)^2}$$

With *pos* the $135 \times 135$ positions in the image, *or* the eight orientations, *ph* the 2 phases of the Gabor filter, and *stimulus* is the preprocessed image and *filter*$_{pos,\ or,\ ph}$ the oriented Gabor filter with a particular position, orientation and phase.

### Spatial summation (step 2)

For each orientation, the oriented contrast energy is then summed across space (for each orientation) using isotropic 2D Gaussian weights (*Figure 4B*):

$$w_{i=(x',y')} = \frac{1}{2\pi\sigma^2}e^{-\frac{(x'-x)^2+(y'-y)^2}{2\sigma^2}}$$

where $w_{i=(x',y')}$ is the weight at position $i$ indexed by coordinates $x'$ and $y'$; $x$ and $y$ indicate the center of the Gaussian; and $\sigma$ indicates the standard deviation of the Gaussian. Note that because of the scaling term, the sum of the weights equals one:

$$\sum_i w_i = 1$$

The first two steps result in eight values (one per orientation) for the oriented contrast energy summed within the pRF defined by the Gaussian.

### Variance (step 3)

A value for the summed oriented contrast energy will be high if an orientation has a high contrast in the area described by the 2D Gaussian and small in case an orientation has low contrast in the area described by the Gaussian. The variance is then calculated across these eight summed oriented contrast energy values, exponentiated with an exponent and multiplied by a gain (*Figure 4C*):

$$response = g \cdot \left( \frac{1}{8} \sum_{k=1}^{8} (x_k - \bar{x})^2 \right)^n$$

With $x_k$ the summed contrast energy or orientation $k$ of which there are a total of 8, $n$ an exponent, $g$ the gain and $\bar{x}$ the mean summed contrast energy. When one orientation is present and others are not (e.g. [1 0 0 0 0 0 0 0]), the variance will be high. When several orientations are present to different degrees (e.g. [.4 .5 .4 .5 .4 .5 .4 .5]), the variance will be low (*Figure 5*). The image contrast and the number of orientations drive the response of the OV model, and the model has five parameters: $x$, $y$, $\sigma$, $g$ and $n$.

## Model fitting

The SOC model was fit to the broadband data and the OV model was fit to narrowband gamma data using leave one out cross validation.

### Fitting broadband power changes with the SOC model

The SOC model (*Figure 4D–G*) was previously developed to explain fMRI signal changes for many of the images that were used here (*Kay et al., 2013b*). To explain ECoG broadband changes with this model, we used a very similar fitting approach.

We fit the SOC model to ECoG broadband response amplitudes from each electrode. Model fitting was performed using nonlinear optimization (MATLAB Optimization Toolbox) with the objective of minimizing squared error. To guard against local minima, we used a variety of initial seeds for the $c$ and $n$ parameters. For every combination of $c$ and $n$, where $c$ is chosen from {.1 .4 .7 .8 .9 .95 1} and $n$ is chosen from {.1 .3 .5 .7 .9 1}, we optimized $x$, $y$, $\sigma$, and $g$ with $c$ and $n$ fixed, and then optimized all of these parameters simultaneously. To optimally fit the pRF location, we first seed the pRF in the center of the stimulus and estimated the model parameters from the SPACE stimuli, and then use the estimated $x$, $y$ and $\sigma$ to fit the model again on all stimuli. To get an unbiased estimate of the model accuracy we fit the model using leave one out cross-validation.

### Fitting narrowband power changes with the new OV model

The OV model has five parameters that need to be estimated: the $x$, $y$, and $\sigma$ of the Gaussian that define the location and size of the population receptive field, an exponent $n$ and a multiplicative gain $g$. The OV model was fit to the ECoG narrowband gamma power changes. The $x$ and $y$ position of the pRF were derived from the SOC model fit to the broadband data from the same electrode. There is a consistent relationship between the eccentricity of the pRF in V1, V2 and V3 and its size (*Kay et al., 2013a*), and we used this relationship to calculate the size $\sigma$ of the pRF. The only parameters that are left to be estimated through fitting are the gain $g$ and the exponent $n$. We tested an exponent of {.1 .2 .3 .4 .5 .6 .7 .8 .9 1} and derived the gain through a linear regression with least squares between the model output and all data except one stimulus. Model performance was then tested on the left out stimuli (leave one out cross-validation).

### Model accuracy

The model performance was evaluated on the data for the left out stimulus (leave one out cross-validation). As a measure for model performance we calculated the coefficient of determination (COD):

$$R^2 = 100 * \left( 1 - \frac{SS_{residuals}}{SS_{data}} \right)$$

$$SS_{residuals} = \sum_i (y_i - f_i)^2$$

$$SS_{data} = \sum_i (y_i)^2$$

where $y_i$ is the measured response amplitude and $f_i$ is the predicted response amplitude for stimulus $i$. Note that $R^2$ is defined here with respect to zero, rather than with respect to the mean response

(similar as in *Kay et al., 2013b*). This metric of prediction accuracy accounts for both accuracy of the mean and the variance across conditions. A model that predicted only the mean (i.e. uses the mean of all other responses as a prediction for the left-out response) would have, averaged across electrodes, a 26% accuracy for gamma oscillations, much lower than the 75% accuracy of the OV model, and 41% for the broadband response, much lower than the 80% accuracy of the SOC model.

## Natural image simulations

We calculated the predictions of the SOC and OV models for a set of natural images. We used a large collection of 771 natural photographs from the McGill Colour Image Database (*Olmos and Kingdom, 2004*). These images were converted to grayscale luminance values using supplied calibration information, cropped to square, and downsampled to 240 × 240 pixels. The images were then further processed and filtered in the exact same way as the stimuli used in the main experiment. Simulated outputs from the SOC and OV models were calculated using the SOC and OV parameters for every electrode. This resulted in 771 simulated SOC and OV responses for each of the 15 electrodes.

## Data availability statement

To foster reproducible research, we make the data and code publicly available via a permanent archive on the Open Science Framework (https://osf.io/eqjxb/ and Github (*Hermes, 2019*; copy archived at https://github.com/elifesciences-publications/Paper_Hermes_2019_eLife). The data provided conform to the Brain Imaging Data Structure (BIDS) (*Gorgolewski et al., 2016*; *Holdgraf et al., 2019*) for ease of use by other researchers.

## Acknowledgements

This work was supported by the Dutch Organization for Scientific Research grant 016.VENI.178.048 to DH and the National Institute of Mental Health grant R01MH111417-01 to JW and NP. The authors thank the Parvizi lab at Stanford, the Stanford Human Intracranial Cognitive Electrophysiology Program (SHICEP), and the Ramsey lab, Cyrille Ferrier and the neurophysiology team at the UMC Utrecht for their help in recording the ECoG data, and we thank David Heeger for helpful discussions and comments on an earlier draft of the manuscript and we thank Pascal Fries for helpful suggestions on an earlier version of this manuscript.

## Additional information

### Funding

| Funder | Grant reference number | Author |
|---|---|---|
| Nederlandse Organisatie voor Wetenschappelijk Onderzoek | 016.VENI.178.048 | Dora Hermes |
| National Institute of Mental Health | R01MH111417-01 | Natalia Petridou Jonathan Winawer |

The funders had no role in study design, data collection and interpretation, or the decision to submit the work for publication.

### Author contributions

Dora Hermes, Conceptualization, Data curation, Software, Formal analysis, Funding acquisition, Visualization, Methodology, Writing—original draft, Writing—review and editing; Natalia Petridou, Conceptualization, Resources, Data curation, Funding acquisition, Writing—review and editing; Kendrick N Kay, Conceptualization, Software, Formal analysis, Visualization, Methodology, Writing—review and editing; Jonathan Winawer, Conceptualization, Resources, Data curation, Software, Formal analysis, Funding acquisition, Methodology, Writing—original draft, Writing—review and editing

## Author ORCIDs

Dora Hermes https://orcid.org/0000-0002-8683-8909
Kendrick N Kay https://orcid.org/0000-0001-6604-9155
Jonathan Winawer https://orcid.org/0000-0001-7475-5586

## Ethics

Human subjects: Subjects gave informed consent and the study was approved by the Institutional Review Board (IRB) at Stanford University (11354) and the Medisch Ethische Toetsingscommissie Utrecht (METC Utrecht) at the UMC Utrecht (14-420) in accordance with the 2013 provisions of the Declaration of Helsinki.

## Decision letter and Author response

Decision letter https://doi.org/10.7554/eLife.47035.031
Author response https://doi.org/10.7554/eLife.47035.032

# Additional files

## Supplementary files

• Transparent reporting form DOI: https://doi.org/10.7554/eLife.47035.025

## Data availability

To foster reproducible research, we make the data and code publicly available via a permanent archive on the Open Science Framework (https://osf.io/eqjxb/) and the code to reproduce the figures can be found on https://github.com/dorahermes/Paper_Hermes_2019_eLife (copy archived at https://github.com/elifesciences-publications/Paper_Hermes_2019_eLife). The data provided conform to the Brain Imaging Data Structure (BIDS) specification (Gorgolewski et al., 2016; Holdgraf et al., 2019) for ease of use by other researchers.

The following dataset was generated:

| Author(s) | Year | Dataset title | Dataset URL | Database and Identifier |
|---|---|---|---|---|
| Hermes D, Winawer J, Kay K | 2019 | GammaOVModel | https://osf.io/eqjxb/ | Open Science Framework, eqjxb |

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
