## [Decision Letter]

**Acceptance summary:**

In this study, Hermes and colleagues develop and test a computational model that predicts oscillatory gamma band responses for various visual stimuli. A narrowband gamma response as measured with intracranial recordings in humans was elicited only by a specific subset of stimuli (i.e., visual gratings), in contrast with the broadband and fMRI responses that were observed for a much wider range of images. The authors suggest that gamma band oscillations might reflect gain control. This study adds to the growing line of research challenging the purported role of gamma oscillations in information communication, at least, in the visual system. Overall this topic is timely, interesting, and sure to generate discussion regarding the mechanistic role of gamma band oscillations. The work is therefore of interest to both cognitive and systems neuroscientists.

**Decision letter after peer review:**

Thank you for submitting your article "An image-computable model for the stimulus selectivity of gamma oscillations" for consideration by *eLife*. Your article has been reviewed by four peer reviewers, including Saskia Haegens as the Reviewing Editor and Reviewer #1, and the evaluation has been overseen by Laura Colgin as the Senior Editor. The following individual involved in review of your submission has agreed to reveal their identity: Hesham ElShafei (Reviewer #3).

The reviewers have discussed the reviews with one another and the Reviewing Editor has drafted this decision to help you prepare a revised submission.

Summary:

In this study, Hermes and colleagues develop (and test) a computational model that predicts gamma responses for various visual stimuli. In comparison to broadband ECoG gamma and fMRI BOLD responses, a narrowband ECoG oscillatory gamma response was elicited by a specific subset of stimuli, i.e., visual gratings. This study adds to the growing line of research challenging the purported role of gamma oscillations in information communication, at least in the visual system.

Essential revisions:

Overall this topic is timely, interesting, and sure to be controversial (in a good way!). Both the experimental part of the study and the modeling part are of high quality in terms of conceptualization, design, implementation and results, and the manuscript is generally well written. That being said, all reviewers agreed that the authors would have to tone down some of the fairly strong/conclusive arguments, connect to the wider literature on gamma, do some more explicit model comparisons, include statistics on the brain data, and separate induced from evoked activity. We detail these suggestions below.

Regarding fairly strong conclusions (i.e., that gamma is a biomarker for gain modulation and cannot support binding/communication/etc):

1) As with all negative findings, one essentially cannot exclude type-2 error, i.e., that the effect was there but the experiment failed to measure it. This is a universally acknowledged limitation of negative findings. Related to this, it would actually be essential that the authors provide statistical significance for the% signal change. The scale of gamma signal change goes up to 2000%, whereas the BOLD only goes up to 5%. It remains unclear how much response-change would be significant, i.e., would a 5% response-change in narrowband gamma response be considered significant or noise?

2) Even if there was no significant% signal change in the gamma band for many stimuli, this still does not mean that gamma oscillations do not play a role in cortical function. See the classic work by the group of Stefano Panzeri on natural images (for instance Belitski et al., 2008) where gamma carries information about the stimulus. Also see more recent work by Besserve et al., 2015, where they show that other observables such as gamma phase convey information transfer. Similarly, there are even reports of neurons locking to gamma oscillations in the absence of visual stimuli (e.g., Vinck et al., 2013). In fact, gamma oscillations have first been described as spontaneous (in the absence of stimuli) oscillations in thalamocortical circuits (see classical work by Steriade et al). Thus, while it is clear that these oscillations are indeed strongly stimulus specific, the fact that not all stimuli here induce large gamma oscillations does not mean that gamma oscillations do not exist or play a general role in cortical function. Furthermore, the authors should discuss how their conclusions fit in with previous reports of narrowband gamma modulation during the anticipation of a stimulus or the maintenance of some of its features. Related, while we are aware this paper deals with gamma oscillations in the visual cortex, it would be relevant to discuss whether the authors believe this (potential) functional role of gamma generalizes to other sensory (e.g., auditory and somatosensory cortices) and non-sensory areas (e.g., prefrontal areas). The general discussion would benefit from such an outlook.

3) This study itself (Figure 9) produces interesting insights into when these oscillations will be prominent, e.g., in grating-like structures where there is some spatial homogeneity (in addition to temporal homogeneity). Thus, there can be more microscopic gamma activity (e.g., at the scale of spiking or LFP receptive fields) in response to image parts that effectively stimulate gamma. The fact that this activity is not strong enough to drive the more macroscopic ECoG signal does not mean that the activity is absent or that other aspects of the activity such as its phase or the locking of neurons to it are not functionally important. What if the dynamic range of gamma -responses is just very high, with high responses for gratings, but with also existing responses for other types of stimuli, which sometimes constitute the majority of the broadband response (e.g., see Electrode 7/8 for curve patterns)?

Regarding the time-windows of analysis:

4) First, why did the authors opt for a baseline interval that could potentially be contaminated by early responses to the images? In addition to pooling all trials to compute this baseline, this might render any conclusion on the magnitude of gamma responses somewhat circular, i.e., weak gamma responses to circular patterns would be even weaker when computed relative to stronger gamma responses to gratings.

5) Evoked responses could potentially confound not only the broadband but also the gamma estimation. The time period that is used for computation of the broadband and gamma response includes the time window (0-200/250ms) of these visual evoked responses. Do the authors differentiate between evoked and induced responses? Do these contribute equally to the narrow- and broadband responses? The inclusion of phase-locking plots would be insightful as these components might reflect different processes.

Regarding model testing and comparison:

6) The SOC model was trained on broadband responses, which also include the narrowband gamma-band, which was used for the OV model. What information is present in activity which does not include the gamma-band? What is the relationship between the two presented models? Can the SOC model be fit to broadband without gamma activity and still yield high R^2^?

7) While the authors show data for all 15 electrodes, the quantification of model output could be improved. While in the natural image simulation study, the models are contrasted against each other explicitly (Figure 9), this is not the case for experimental work. Currently only the mean R^2^ is reported for each model, not the dispersion. That is, the authors should perform explicit model comparisons: run the SOC model on the gamma response and the OV model on the broadband response and show that the SOC model predicts broadband better than gamma and vice versa. Additionally, it is not clear a straight linear regression is the best approach here; what do the fits look like if a non-parametric Spearman correlation is used?

8) "Note that R^2^ is defined here with respect to zero, rather than with respect to the mean response (similar as in (Kay et al., 2013b)…" This needs to be elaborated as this is crucial for the conclusions. Furthermore, it would be good to see some statistical significance of the accuracies with respect to chance level.

---

## [Author Response]

Essential revisions:Overall this topic is timely, interesting, and sure to be controversial (in a good way!). Both the experimental part of the study and the modeling part are of high quality in terms of conceptualization, design, implementation and results, and the manuscript is generally well written. That being said, all reviewers agreed that the authors would have to tone down some of the fairly strong/conclusive arguments, connect to the wider literature on gamma, do some more explicit model comparisons, include statistics on the brain data, and separate induced from evoked activity. We detail these suggestions below.

We thank the reviewers for their positive comments. We have addressed the issues regarding tone and relationship to the literature, we have added model comparisons, and we have included statistics and separated induced from evoked activity; see details below. (Note that we have added numbers to the reviewer comments for organizational purposes.)

Regarding fairly strong conclusions (i.e., that gamma is a biomarker for gain modulation and cannot support binding/communication/etc):

Regarding the issue of the strength of the stated claims in the paper, we have made several modifications:

We have clarified that the idea that gamma oscillations might be a biomarker for gain modulation is an interpretation, not a proven conclusion. Because our results provide a new model that makes accurate predictions for responses to many visual stimuli, we think it is our responsibility as authors to offer possible interpretations. Gain modulation seems most likely to us. We also now highlight the fact that others have also suggested the idea that gamma oscillations reflect gain control; thus, this theory does not rest solely on our data. (subsection “Gamma oscillations and gain control”, third paragraph).

We also clarify the fact that gamma oscillations *could* certainly support other functions such as long-range communication. We note that a theory proposing these oscillations as the principal means of long-range visual communication would have to also account for our findings (namely that the measured oscillations are small or even within the noise level for many stimuli; see also response to point 1 regarding the possibility of undetected signals). (subsection “Relevance for neuronal and cognitive function”, second and last paragraphs).

We have made several other relevant textual changes including (Results, second paragraph; subsection “Gamma oscillations and gain control”, first paragraph; subsection “Conclusions”).

We believe that these modifications help contextualize, clarify, and temper the strength of our claims.

1) As with all negative findings, one essentially cannot exclude type-2 error, i.e., that the effect was there but the experiment failed to measure it. This is a universally acknowledged limitation of negative findings. Related to this, it would actually be essential that the authors provide statistical significance for the% signal change. The scale of gamma signal change goes up to 2000%, whereas the BOLD only goes up to 5%. It remains unclear how much response-change would be significant, i.e., would a 5% response-change in narrowband gamma response be considered significant or noise?

We agree that it is impossible to definitively prove that an effect is identical to zero. However, in this study, we have measured broadband and gamma signals with fairly high reliability. Thus, we can put confidence bounds on the magnitude of broadband and gamma signals, and can make reasonable inferences about the sizes of these signals. In other words, although we cannot prove that gamma signals are zero (for certain stimuli, see subsection “Relevance for neuronal and cognitive function”, second paragraph), we can be confident that the magnitude of gamma signals (for certain stimuli) is either zero or quite small.

To clarify the inferences that we have made in this study, we now have included more extensive discussion about how to interpret the amplitude of changes in different signals (subsection “Gamma responses are well predicted by a model that is sensitive to variation in orientation content”, second paragraph; subsection “Different origins of broadband and narrowband gamma”, last paragraph). We also now perform statistical tests (as suggested by the reviewers) comparing the measured responses to the level of background noise. This includes changes to Figures 2, 3, 6 and legends, Figure 3—figure supplement 1-2, Figure 6—figure supplement 1-2 and changes throughout the Results and in the Materials and methods.

2) Even if there was no significant% signal change in the gamma band for many stimuli, this still does not mean that gamma oscillations do not play a role in cortical function. See the classic work by the group of Stefano Panzeri on natural images (for instance Belitski et al., 2008) where gamma carries information about the stimulus. Also see more recent work by Besserve et al., 2015, where they show that other observables such as gamma phase convey information transfer. Similarly, there are even reports of neurons locking to gamma oscillations in the absence of visual stimuli (e.g., Vinck et al., 2013). In fact, gamma oscillations have first been described as spontaneous (in the absence of stimuli) oscillations in thalamocortical circuits (see classical work by Steriade et al). Thus, while it is clear that these oscillations are indeed strongly stimulus specific, the fact that not all stimuli here induce large gamma oscillations does not mean that gamma oscillations do not exist or play a general role in cortical function.

We agree that the oscillations, when they are observed, might conceivably play any number of roles in vision, cognition, and neuronal communication. For theories that posit such important roles, our study may be particularly interesting, because we find that the level of the oscillations varies in ways not easily predicted by such theories, and even that for some stimuli, given our measurement and analytic tools, there are no measurable oscillations beyond the baseline noise. In addition, some studies which report “gamma-band” activity may be measuring the portion of the broadband response that happens to overlap the gamma band (~50-80 Hz). To acknowledge and contextualize these issues, we have modified the paper accordingly. (Results, first paragraph; subsection “Different origins of broadband and narrowband gamma”, second and last paragraphs; subsection “Relevance for neuronal and cognitive function”)

Furthermore, the authors should discuss how their conclusions fit in with previous reports of narrowband gamma modulation during the anticipation of a stimulus or the maintenance of some of its features. Related, while we are aware this paper deals with gamma oscillations in the visual cortex, it would be relevant to discuss whether the authors believe this (potential) functional role of gamma generalizes to other sensory (e.g., auditory and somatosensory cortices) and non-sensory areas (e.g., prefrontal areas). The general discussion would benefit from such an outlook.

We now discuss the relationship between our measurements and other types of observed gamma oscillations (subsection “Stimulus selectivity of gamma oscillations in visual cortex”, last paragraph).

3) This study itself (Figure 9) produces interesting insights into when these oscillations will be prominent, e.g., in grating-like structures where there is some spatial homogeneity (in addition to temporal homogeneity). Thus, there can be more microscopic gamma activity (e.g., at the scale of spiking or LFP receptive fields) in response to image parts that effectively stimulate gamma. The fact that this activity is not strong enough to drive the more macroscopic ECoG signal does not mean that the activity is absent or that other aspects of the activity such as its phase or the locking of neurons to it are not functionally important. What if the dynamic range of gamma-responses is just very high, with high responses for gratings, but with also existing responses for other types of stimuli, which sometimes constitute the majority of the broadband response (e.g., see Electrode 7/8 for curve patterns)?

We address the issue of dynamic range with new analyses comparing all measurement types to background variability (see response to Point 1 above). We also explicitly discuss the possibility of missing small oscillations due to the measurement method(subsection “Relevance for neuronal and cognitive function”, second paragraph).

Regarding the time-windows of analysis:4) First, why did the authors opt for a baseline interval that could potentially be contaminated by early responses to the images? In addition to pooling all trials to compute this baseline, this might render any conclusion on the magnitude of gamma responses somewhat circular, i.e., weak gamma responses to circular patterns would be even weaker when computed relative to stronger gamma responses to gratings.

We are not sure what is meant by early responses to the images and, just to clarify, we do have periods between the images where there is nothing on the screen. We have clarified this in the manuscript (subsection “Stimuli and task”): each stimulus was shown for 500 ms followed by a gray screen for 500 ms. The baseline period consists of the 500 ms gray screen period, during which there is no stimulus on the screen.

We also now calculate the gamma and broadband responses during the baseline period (subsection “Spectral analysis”, first paragraph, subsection “Bootstrapping and confidence intervals”) and test for each stimulus whether the response is significantly greater than baseline (subsection “Significance testing”) and we add this with blue dots in Figures 2, 3 and Figure 3—figure supplements 1-3 and Figure 6—figure supplements 1-3.

In addition, one can be concerned about late responses from the previous image bleeding into the baseline. As shown in the response to issue 6, we also calculate the responses and statistics for the 200-500 ms after stimulus and baseline onset. This window does not change the results and we have added this to the Figure 3—figure supplement 3 and Figure 6—figure supplement 3.

5) Evoked responses could potentially confound not only the broadband but also the gamma estimation. The time period that is used for computation of the broadband and gamma response includes the time window (0-200/250ms) of these visual evoked responses. Do the authors differentiate between evoked and induced responses? Do these contribute equally to the narrow- and broadband responses? The inclusion of phase-locking plots would be insightful as these components might reflect different processes.

The goal of our paper is to investigate induced changes to spectral power in ECoG signals. We have now performed the following controls to ensure that evoked responses do not form the major contribution to the reported broadband and gamma power changes. First, we calculate phase-locking plots (new figure supplement: Figure 2—figure supplement 1). This figure shows that the evoked responses carry most power in the lower frequencies. Second, we calculate broadband and gamma responses (as before (Hermes et al., 2015)) in the timeframe after the major visual evoked response (200-500 ms) and show similar broadband and gamma power changes across stimuli as in the 0-500 ms window (Figure 3—figure supplement 3 and Figure 6—figure supplement 3). This indicates that evoked responses are not a major contributor to the spectral power changes. We have noted this in the subsection “Spectral analysis”.

Regarding model testing and comparison:6) The SOC model was trained on broadband responses, which also include the narrowband gamma-band, which was used for the OV model. What information is present in activity which does not include the gamma-band? What is the relationship between the two presented models? Can the SOC model be fit to broadband without gamma activity and still yield high R^2^?

We apologize that the manuscript was not clear regarding the nature of how broadband and narrowband responses are quantified. To clarify, the magnitude of broadband responses and the magnitude of narrowband responses on a given trial are, from the point of view of the analysis, distinct and independent quantities (i.e. there is no constraint that induces a relationship between the two). If, in experimental data, the two quantities turn out to be related (e.g. correlated), this is an empirical finding.

We have revised the paper to clarify the relationship between the broadband and narrowband signals and discuss their independence (Results, first paragraph; subsection “Spectral analysis”). In addition, as suggested by the reviewer, we now fit both models (SOC and OV) to both signal types (narrowband gamma and broadband); the results are shown in a new figure (Figure 7) and table (Table 1) and discussed in the text (subsection “Gamma responses are well predicted by a model that is sensitive to variation in orientation content”, last paragraph). In short, the SOC model does not yield satisfactory fits to gamma responses. Finally, we clarify the relationship between the two models in the text (subsection “An image-computable model of narrowband gamma responses”, second paragraph, subsection “The orientation-variance (OV) model of gamma responses”, first paragraph) and with a new figure panel illustrating the SOC model in Figure 4.

7) While the authors show data for all 15 electrodes, the quantification of model output could be improved. While in the natural image simulation study, the models are contrasted against each other explicitly (Figure 9), this is not the case for experimental work. Currently only the mean R^2^ is reported for each model, not the dispersion. That is, the authors should perform explicit model comparisons: run the SOC model on the gamma response and the OV model on the broadband response and show that the SOC model predicts broadband better than gamma and vice versa. Additionally, it is not clear a straight linear regression is the best approach here; what do the fits look like if a non-parametric Spearman correlation is used?

Here, the reviewer requests more thorough quantification and comparison of model performance. Accordingly, we have now added explicit model comparisons (see also response to point 5 above) and also a comparison between each of our main models (OV and SOC) against a model that predicts the same level of response to all stimuli (i.e. a mean model). Please see new Figure 7 and new Table 1, as well as related text (subsection “Gamma responses are well predicted by a model that is sensitive to variation in orientation content”, last paragraph). The results show clearly that, as predicted by the reviewer, the SOC model is more accurate for broadband, while the OV model is more accurate for gamma.

The reviewer also asks what the results from a Spearman rank test would look like. Although we can calculate a Spearman rank correlation, we do not think that this would represent the results well for several reasons. Firstly, the two models are not cases of simple linear regression, but reflect nonlinear parameter optimization. Each model includes an exponent *n*, which captures some of the non-linearities in the data. The Spearman rank correlation as a model performance measure would be insensitive to whether the exponent was fit well or not. Similarly, each model includes a gain *g*, which would be ignored when using the Spearman rank correlation to quantify model performance.

Secondly, our models predict the actual response amplitude, not merely the rank of the images. If there are only a few images that elicit a reliable large response, Spearman will not give an accurate picture of the model fit. To demonstrate, we simulate the following 1000 times (Author response image 2): we simulate responses for 50 stimuli: 45 just elicit noise, and 5 drive an increasingly larger response. These 5 responses are not outliers, but reliable data points that are well predicted. While 5 images are in the correct rank order, the rank order of the other 45 images is going to be random. This unduly reduces the size of a coefficient of determination calculated based on rank rather than amplitude. See Author response image 1, Matlab code snippet.

**Author response image 1. respfig1:** Matlab code.

**Author response image 2. respfig2:** Output to simulate rank order statistic. We simulated an experiment 1,000 times. In each case, there are 5 stimuli that drive systematic responses (amplitudes of 1, 2, 3, 4, and 5) and 45 stimuli that evoke responses centered at 0 (left panel). The scatter plot shows that the model fit is good. This is reflected by a high coefficient of determination (‘cod data’, blue); however, based on the rank ordered data points, the cod is poor, usually negative (‘cod ranks’, red).

We can calculate a squared Spearman correlation on our data and model predictions, and the results from this analysis are shown in Author response image 3. It is still clear that the OV model performs best in predicting gamma and the SOC model performs best in predicting broadband, but the actual size of the prediction is heavily influenced by the number of images that result in a reliable response, as expected from the simulation shown in Author response image 2. We believe that this produces results that are more difficult to interpret than a more straightforward measure that does not convert data to ranks.

**Author response image 3. respfig3:** OV and SOC model performance on broadband and gamma power. In this plot, model performance was calculated by the squared Spearman correlation. The x-axis shows the model performance of the OV model fit to the gamma power (red) and broadband power (blue). The y-axis shows the model performance of the SOC model fit to the gamma power (red) and broadband power (blue). The small dots show the performance for individual electrodes and the large dot indicates the mean +/- two times the standard error.

8) "Note that R^2^ is defined here with respect to zero, rather than with respect to the mean response (similar as in (Kay et al., 2013b).…" This needs to be elaborated as this is crucial for the conclusions. Furthermore, it would be good to see some statistical significance of the accuracies with respect to chance level.

Using deviations compared to zero, rather than compared to the mean avoids the arbitrariness of the mean, which varies from electrode to electrode and is part of what the model attempts to fit. To calculate a ‘chance level’ as suggested by the reviewer, we have now calculated how well the models capture the data compared to a mean prediction, that is, a model that predicts the same response level to each stimulus (see also point 6). The OV and SOC model both perform significantly better than the mean prediction (p<0.001), indicating that these models successfully capture variance in the data beyond the mean response level. See subsection “Gamma responses are well predicted by a model that is sensitive to variation in orientation content”, last paragraph, subsection “Model Accuracy” and Table 1.